# Surface-to-space atmospheric waves from Hunga Tonga–Hunga Ha'apai eruption

Corwin J. Wright[1✉], Neil P. Hindley[1], M. Joan Alexander[2], Mathew Barlow[3], Lars Hoffmann[4], Cathryn N. Mitchell[1], Fred Prata[5,6], Marie Bouillon[7], Justin Carstens[8], Cathy Clerbaux[7], Scott M. Osprey[9], Nick Powell[10], Cora E. Randall[11,12] & Jia Yue[13,14]

The January 2022 Hunga Tonga–Hunga Ha'apai eruption was one of the most explosive volcanic events of the modern era[1,2], producing a vertical plume that peaked more than 50 km above the Earth[3]. The initial explosion and subsequent plume triggered atmospheric waves that propagated around the world multiple times[4]. A global-scale wave response of this magnitude from a single source has not previously been observed. Here we show the details of this response, using a comprehensive set of satellite and ground-based observations to quantify it from surface to ionosphere. A broad spectrum of waves was triggered by the initial explosion, including Lamb waves[5,6] propagating at phase speeds of $318.2 \pm 6$ m s$^{-1}$ at surface level and between $308 \pm 5$ to $319 \pm 4$ m s$^{-1}$ in the stratosphere, and gravity waves[7] propagating at $238 \pm 3$ to $269 \pm 3$ m s$^{-1}$ in the stratosphere. Gravity waves at sub-ionospheric heights have not previously been observed propagating at this speed or over the whole Earth from a single source[8,9]. Latent heat release from the plume remained the most significant individual gravity wave source worldwide for more than 12 h, producing circular wavefronts visible across the Pacific basin in satellite observations. A single source dominating such a large region is also unique in the observational record. The Hunga Tonga eruption represents a key natural experiment in how the atmosphere responds to a sudden point-source-driven state change, which will be of use for improving weather and climate models.

On 15 January 2022, the Hunga Tonga–Hunga Ha'apai submarine volcano (20.54° S, 175.38° W, hereafter 'Hunga Tonga') erupted, producing a vertical plume more than 30 km tall with overshooting tops above 55 km, which is a record in the satellite era[3] and probably longer[2]. From surface pressure data, we estimate a single-event energy release from the initial explosion of between 10 and 28 EJ, which is probably larger than the 1991 Mt Pinatubo eruption[2] (around 10 EJ), and possibly comparable to Krakatoa in 1883 (ref. [2]; around 30 EJ) (Methods and Extended Data Fig. 1a,b).

Large explosions such as volcanoes and nuclear tests are theoretically understood to produce atmospheric waves[10,11] across a range of length and frequency scales. At short horizontal wavelengths, these include external Lamb waves[5,6,12], acoustic waves[11] and internal gravity waves[13]. In addition to explosion-generated waves, volcanoes can also act as a sustained wave source after the initial eruption through updraughts and heating associated with plume convection[14,15].

In practice, observations of such waves at subacoustic frequencies after volcanic eruptions are rare. Krakatoa[6] and Pinatubo[16], among others, produced strong Lamb waves visible in surface pressure. Internal

waves in the boundary layer have been inferred from seismography, barometry and infrasound for eruptions including El Chichon[14] (1982), Pinatubo[14] and Okmok[15] (2008). In the free atmosphere, local gravity wave activity associated with plume convection has been seen in mesospheric nightglow over the La Soufrière[17] (2021) and Calbuco[13] (2015) eruptions and in local cloud over eruptions including Cumbre Vieja (2021). Re-examination of 1990s Advanced Very High Resolution Radiometer data also shows waves in cloud above Pinatubo (Extended Data Fig. 2). Finally, an electron-density ionospheric wave response is usually observed[18–22], with the response magnitude proposed as a metric of volcano explosive power[23].

There is, however, no direct observational evidence for long-distance propagation in the free electrically neutral atmosphere of either Lamb or gravity waves triggered by volcanoes. Pre-2000s satellite observations had insufficient resolution and coverage to measure such waves, and no event since[8] has produced a wave response similar to that identified within hours[24] of Hunga Tonga. This eruption thus represents an opportunity to quantify the wave response to a point-source disruption at a scale and comprehensiveness unique in the observational record.

[1]Centre for Space, Atmospheric and Oceanic Science, University of Bath, Bath, UK. [2]Northwest Research Associates, Boulder, CO, USA. [3]Environmental, Earth & Atmospheric Sciences, University of Massachusetts Lowell, Lowell, MA, USA. [4]Jülich Supercomputing Center, Forschungszentrum Jülich, Jülich, Germany. [5]AIRES Pty Ltd, Mt Eliza, VIC, Australia. [6]School of Electrical Engineering, Computing & Mathematical Science, Curtin University, Bentley, WA, Australia. [7]LATMOS/IPSL, Sorbonne Université, UVSQ, CNRS, Paris, France. [8]Center for Space Science and Engineering Research, Bradley Department of Electrical and Computer Engineering, Virginia Tech, Blacksburg, VA, USA. [9]Atmospheric, Oceanic and Planetary Physics, Department of Physics, University of Oxford, Oxford, UK. [10]Raytheon Technologies, Waltham, MA, USA. [11]Laboratory for Atmospheric and Space Physics, University of Colorado Boulder, Boulder, CO, USA. [12]Department of Atmospheric and Oceanic Sciences, University of Colorado Boulder, Boulder, CO, USA. [13]NASA Goddard Space Flight Center, Community Coordinated Modeling Center, Greenbelt, MD, USA. [14]Physics Department, Catholic University of America, Washington, DC, USA. ✉e-mail: c.wright@bath.ac.uk

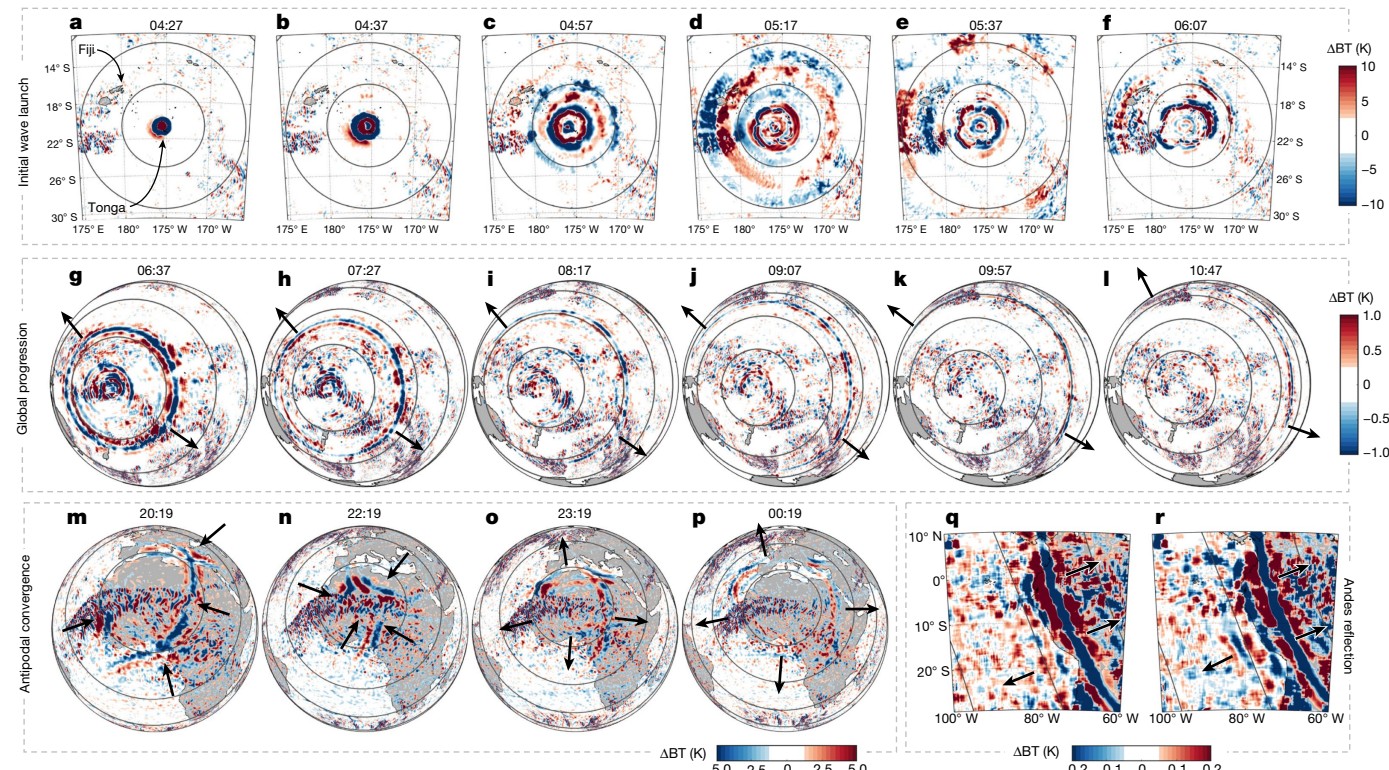

**Fig. 1 | Initial Lamb wave propagation in the troposphere.** Brightness temperature changes (ΔBT) observed by GOES (**a**–**l**), the Meteosat Spinning Enhanced Visible and InfraRed Imager (SEVIRI) (**m**–**p**) and GOES-EAST (**q**,**r**). Range rings indicate distance from Hunga Tonga in 500 km (**a**–**f**) and 2,000 km (**g**–**r**) steps. To reduce noise from weather systems, global and antipodal panels have been processed with a 200-km-radius Wiener filter, and Andes panels with a 400 km boxcar and 72-km-radius Wiener filter. Black arrows indicate approximate wave location and propagation direction. All times are UTC.

## Eruption and immediate wave response

Figures 1 and 2 show the propagation of Lamb and gravity waves triggered by the initial eruption on 15 January 2022; Fig. 1 shows height-integrated data from the Geostationary Operational Environmental Satellite (GOES) and Meteosat platforms and Fig. 2 shows height-resolved measurements from multiple instrument types in addition to GOES.

The eruption became visible just before 04:00 Coordinated Universal Time (UTC) as a plume that reached a width of 200 km and height of more than 30 km within 30 min (ref. [3]). Then, 20–30 min after the plume began rising, an atmospheric wave became visible in 10-min-resolution near-infrared geostationary imagery. Back-projection from surface pressure data shows that the trigger source occurred at 04:28 ± 0:02 UTC, with the leading wavefront propagating away at a near-surface phase speed of 318.2 ± 6 m s$^{-1}$ (Fig. 2c, Extended Data Fig. 1c,d and Supplementary Video 1). On the basis of the high phase speed, large amplitude and non-dispersive nature of the signal, we identify this as a Lamb wave. This type of wave is non-dispersive, and the observed speed is consistent with the Lamb wave produced by Krakatoa, estimated[25] to have propagated at 318.8 ± 3 m s$^{-1}$.

The Hunga Tonga Lamb wave propagated around the globe, passing through the antipodal point in Algeria 18.1 h (±7.5 min) after the eruption (Fig. 1). By this time, the wavefront had deformed because of atmospheric and surface processes, and passed through the antipode as four distinct wavefronts (Fig. 1m–p). Over the following days, it was tracked propagating at least three times[4,26] around the Earth. We also see a faint signal in GOES data consistent with the wave being partially reflected from the Andes on its first transit (Fig. 1), and evidence of the wave being slowed over South America (Extended Data Fig. 10).

Using radiance data from the Atmospheric Infrared Sounder (AIRS), Cross-track Infrared Sounder (CrIS) and Infrared Atmospheric Sounding Interferometer (IASI) polar-orbiting thermal infrared (IR) sounders (specifically, 4.3 μm data sensitive to altitudes approximately 39 ± 5 km and 15 μm data sensitive to both the approximately 25 ± 5 km and 42 ± 5 km altitude levels separately, Fig. 2a), we see the Lamb wave as a large-amplitude monochromatic pulse with a phase speed of between 308 ± 5 and 319 ± 4 m s$^{-1}$ depending on the location. We also observe it as a pulse just above the noise floor of Cloud Imaging and Particle Size (CIPS) Rayleigh albedo anomaly data 12,300 km away from and 10.75 h after the eruption (around 55 ± 5 km altitude, phase speed 316–319 m s$^{-1}$, Extended Data Fig. 4a), and as phase fronts in hydroxyl airglow over Hawai'i, 4,960 km away from and 4.3 h after (approximately 87 ± 4 km altitude, phase speed 318 m s$^{-1}$).

The observed Lamb wave phase fronts are uniform in height and phase speed to within the error range of each instrument from the surface to at least the upper mesosphere/lower thermosphere. The energy density of a Lamb wave is theoretically expected[27] to decay exponentially with height, and the observed phase speed is consistent with a vertical mean of sound speed weighted according to this energy distribution (Methods). Our data may show evidence of a slightly different speed for propagation in different directions across the Earth (for example, at Broome, Australia, we measure 319 m s$^{-1}$ for the westward-travelling wave and 316 m s$^{-1}$ for the eastward, Extended Data Fig. 1e), but this is within the uncertainty range of our measurements. The asymmetric perturbations we observe are consistent in sign with such a shift due to background winds.

Following the Lamb wave, we observe a series of slower waves with continually varying speeds and horizontal wavelengths ($\lambda_h$) that we identify as a dispersive packet of fast internal gravity waves (Fig. 2a). These have phase speeds between 240 and 270 m s$^{-1}$, varying with local $\lambda_h$. The leading phase front has the largest amplitude and longest $\lambda_h$, with a brightness temperature (BT) amplitude of 0.74 K and $\lambda_h$

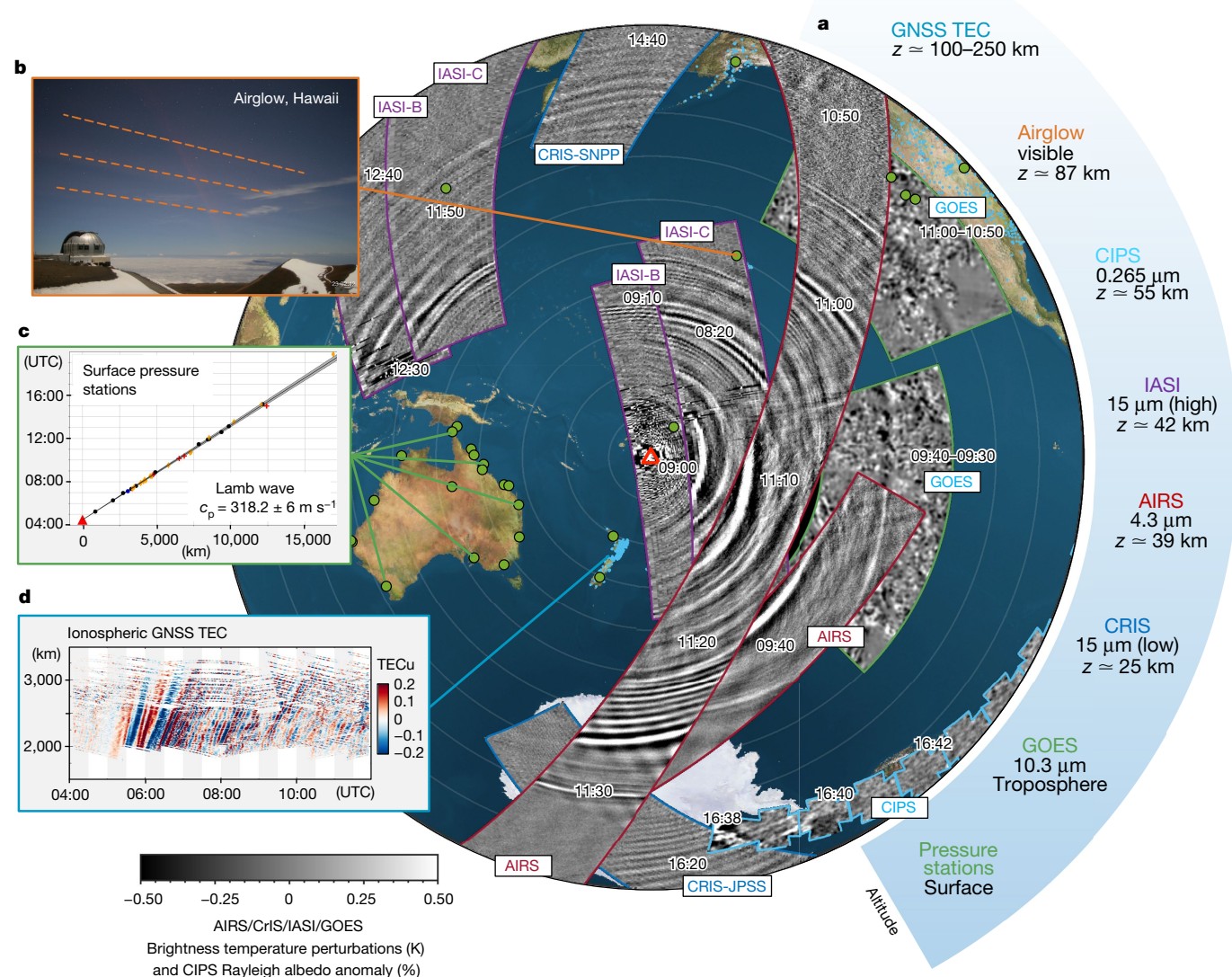

**Fig. 2 | Initial gravity wave and Lamb wave propagation at all heights.**
**a**, Combined measurements of the initial wave release as observed by multiple platforms, listed with their approximate altitudes at right and at times as indicated by overlaid text labels. **c,d**, Pressure (**c**) and TEC (**d**) distance/time series are reproduced as Extended Data Figs. 1d and 3, respectively. Note that AIRS, CrIS and IASI all measure the same three stratospheric altitude channels, but only one is used here from each instrument to show all levels while maintaining visual clarity; owing to the long vertical wavelengths of the observed waves, all three levels are near-identical. **b**, A northward view containing the Lamb wavefront at 09:20 UTC, around 30 min after the wave passed overhead. $c_p$, phase speed; GNSS, global navigation satellite system; JPSS, Joint Polar Satellite System; SNPP, Suomi National Polar-orbiting Partnership. Airglow image: NSF NoirLab.

of 380 km falling to 0.15 K and 100 km across the packet width. This packet is observed to extend approximately 2,000 km and eight phase cycles are visible across the South Pacific around 7 h after generation (Extended Data Fig. 5). We observe the packet over multiple orbits of AIRS, CrIS and IASI across the globe, in CIPS over Antarctica and in airglow (approximately 85 km altitude and depth 8 km) above Hawaiʻi. Vertical wavelength ($\lambda_z$) is poorly defined but very deep: no phase difference is seen between AIRS observations at 25 and 42 km altitude, and calculations based on observed speed and $\lambda_h$ imply that $\lambda_z \gg 110$ km, that is, it is greater than the depth of the homosphere. These phase speeds are consistent with vertically propagating gravity waves travelling at speeds close to, but very slightly less than, the theoretical maximum speeds achievable before total internal reflection (Methods and Extended Data Fig. 6) and with the same temporal origin and source as the Lamb wave.

This leading gravity wave packet passes through the antipode at times between around 00:30 and 02:30 UTC on 16 January 2022, that is 20–22 h after the eruption (Extended Data Fig. 7a–c), with the broad time window determined by separation of different $\lambda_h$ components with time. Gravity waves remaining coherent and expanding over the whole globe from a single source of any kind are unprecedented in the observational record[8]. On their return journey from the antipode, the waves become difficult to distinguish in our intermittent low-Earth orbit satellite snapshots from those produced both later by Hunga Tonga and by other sources, and consequently we cannot track them to their extinction.

The gap between the initial Lamb wave and subsequent gravity wave grows with time. This is consistent with a theoretically predicted forbidden phase speed range between external Lamb wave and internal gravity wave limits imposed by total internal reflection (Extended Data Fig. 6).

Two smaller-amplitude wavefronts are present in the gap; these propagate with the same speed as the leading Lamb wavefront, but trace back to different origin times (Fig. 2a and Extended Data Fig. 4b). We therefore identify these as Lamb waves triggered by subsequent smaller explosions, which were also observed in local surface pressure (Extended Data Fig. 8).

Ionospheric data (Fig. 2d and Extended Data Fig. 3) show key differences from the lower atmosphere. Over New Zealand, we see three large travelling ionospheric disturbances (TIDs), with phase speeds, $\lambda_h$ and amplitudes of (1) 667 m s$^{-1}$, 1,000 km, 0.1 total electron content units (TECu); (2) 414 m s$^{-1}$, 700 km, 0.4 TECu and (3) 343 m s$^{-1}$, 400 km and >0.3 TECu, respectively. The speed and propagation direction of these waves is consistent with a Hunga Tongan source between 04:15 and 05:00, but they do not share the arrival time, phase speed or $\lambda_h$ of the Lamb wave in other atmospheric layers. Therefore, we do not identify these TIDs as the Lamb wave. However, a strong and brief total electron content (TEC) modulation, spiking at an amplitude of more than 0.6 TECu, is seen at 06:15, which is consistent with the expected arrival time and brief period of the Lamb wave.

We do not see TID 1 over North America, but do see a signal consistent with TID 2 and another TID (4) with phase speed around 311 m s$^{-1}$, which is consistent with a later surface pressure perturbation measured over Tonga. We again see a strong TEC modulation at the expected Lamb wave arrival time.

The properties of TIDs 1 and 2 are inconsistent with slant path gravity waves propagating from Hunga Tonga, but these TIDs could have reached the observed sites by indirect paths, for example by vertically propagating as acoustic or gravity waves above the volcano then travelling at high horizontal speeds through the ionosphere. The properties of TIDs 3 and 4 are consistent with the wave activity generated over Hunga Tonga in the hours after the primary eruption.

## Sustained post-eruption wave generation

After the initial trigger, sustained gravity wave generation is seen in the clouds above Hunga Tonga and radiating outwards across the Pacific basin. Although smaller in amplitude and slower in phase speed than those from the initial eruption, these waves are also highly anomalous relative to past gravity wave observations.

Figure 3 shows BT measurements from the GOES 10.3 μm channel over the Hunga Tonga area (Fig. 3a–d) and the AIRS, CrIS and IASI 4.3 μm stratospheric channels over the Pacific basin (Fig. 3e–g) for selected times.

In GOES observations of the eruption cloud top (Fig. 3a–c and Supplementary Video 2), arced features consistent in morphology and temporal progression with propagating concentric gravity wave phase fronts are visible. The value of $\lambda_h$ ranges from the 3 km resolution limit of the data to 65 km, and the BT amplitude from 0.5 to 8 K. These measured properties are very similar to those of gravity waves generated near the convective centres of hurricanes.

The apparent centre of these waves is slightly west of Hunga Tonga. This is consistent with refraction of the wave field by the prevailing easterly winds. The waves are notably consistent in concentric shape over several hours, suggesting a powerful and relatively persistent pulsing source for wave generation. The source may be pulses of convection within the plume above the volcano. The waves weaken in amplitude over time, particularly after 15:00 UTC, but are visible until at least 19:20 UTC (Fig. 3d). They are not found on subsequent days. These results suggest that the volcano may have created a sustained source of convectively generated waves for nearly 15 h after the initial eruption.

Stratospheric AIRS, CrIS and IASI observations (Fig. 3e–g and Extended Data Fig. 7d–o) show wave activity across a range of spatial, frequency and amplitude scales throughout the Pacific basin, all centred on Hunga Tonga. Tracking individual phase fronts is challenging as these data are near-instantaneous at any given location, but conservatively the distribution must include waves with phase speeds of more

than 100 m s$^{-1}$. For example, small-scale continuous wavefronts centred on Hunga Tonga are clearly visible near Japan before 16:00 in Fig. 3g and, even if emitted at the earliest possible time of 04.28 UTC, must have phase speeds around 200 m s$^{-1}$ to have travelled this far. Unlike more typical observed waves, these waves can therefore propagate with little apparent influence from global wind patterns because of their unusually large phase speeds. Such fast speeds reduce normal dissipation effects, enabling the waves to propagate vast distances and affect much higher altitudes than typical gravity waves.

These waves dominate the stratospheric gravity wave spectrum over a radius more than 9,000 km for over 12 h (Extended Data Fig. 7d–o). This is exceptional for a single source and unique in our observational record[8,9]. Orographic wave sources often persist for longer, but are spatially localized; although some waves in the southern polar jet may have propagated downstream[28,29] or laterally[8,30] from orographic sources, the area they affect is an order of magnitude smaller than here and the waves themselves are highly intermittent. Waves from non-orographic sources such as tropical convection and extreme events such as hurricanes, meanwhile, typically become indistinguishable from background within 2000–3500 km (refs. [31,32]).

## How were the waves generated?

Although we cannot directly observe the generation of the waves owing to insufficient temporal resolution (for the initial explosion) and ash plume blocking effects (for both the initial explosion and subsequent wave generation), the observed wave properties and context allow us to infer the likely mechanisms by which they were generated.

The strong initial response is probably due to the eruption's shallow submarine context and large explosive power. As the volcanic vent was only tens to hundreds of metres below the water[33], the seawater did not suppress the blast but was instead flash-boiled[34] and propelled into the stratosphere. Here it condensed, releasing latent heat near-instantaneously across a depth of tens of kilometres. This strong and short-lived forcing would produce vertically deep waves across a broad spectrum, consistent with observations. This mechanism is also consistent with significant and large IASI-observed increases in stratospheric water vapour (Extended Data Fig. 9) and $H_2SO_4$ in the plume relative to what would be expected for an eruption of this size. This is in turn consistent with the speculation that, owing to insufficient volcanogenic $SO_2$ and the time available to produce $H_2SO_4$ from $SO_2$, the observed $H_2SO_4$ was formed from $SO_4^{2-}$ released from seawater.

Subsequent wave generation is probably due to similar processes as standard convective waves, such as mechanical oscillator effects[35] associated with vertical air motion within the plume or pulsing from the volcanic heat source below. Such forces would produce sufficiently strong perturbations to generate gravity waves visible both in the plume and propagating freely away. Such a mechanism is again consistent with our observations, particularly the similarity in morphology and amplitude of the observed waves to the concentric generated by hurricanes[36,37] and convective weather systems[32,38].

Another possibility is that the eruptive energy could have transferred to tsunami waves and the tsunamis in turn have generated the waves we observe[39]. However, we argue that this is less likely than simple linear propagation from a convective atmospheric source owing to the highly regular concentric nature of the observed atmosphere waves in Fig. 3e–g, which show no significant evidence of tsunami deformation effects. Other studies have shown that the atmospheric waves also generated meteotsunamis in both the Pacific and other basins[40,41], highlighting the complex interplay between ocean and atmospheric waves in the Earth system.

## Weather/climate forecasting implications

Even though in recent years we have been able to routinely characterize gravity waves in observational data, understanding how the

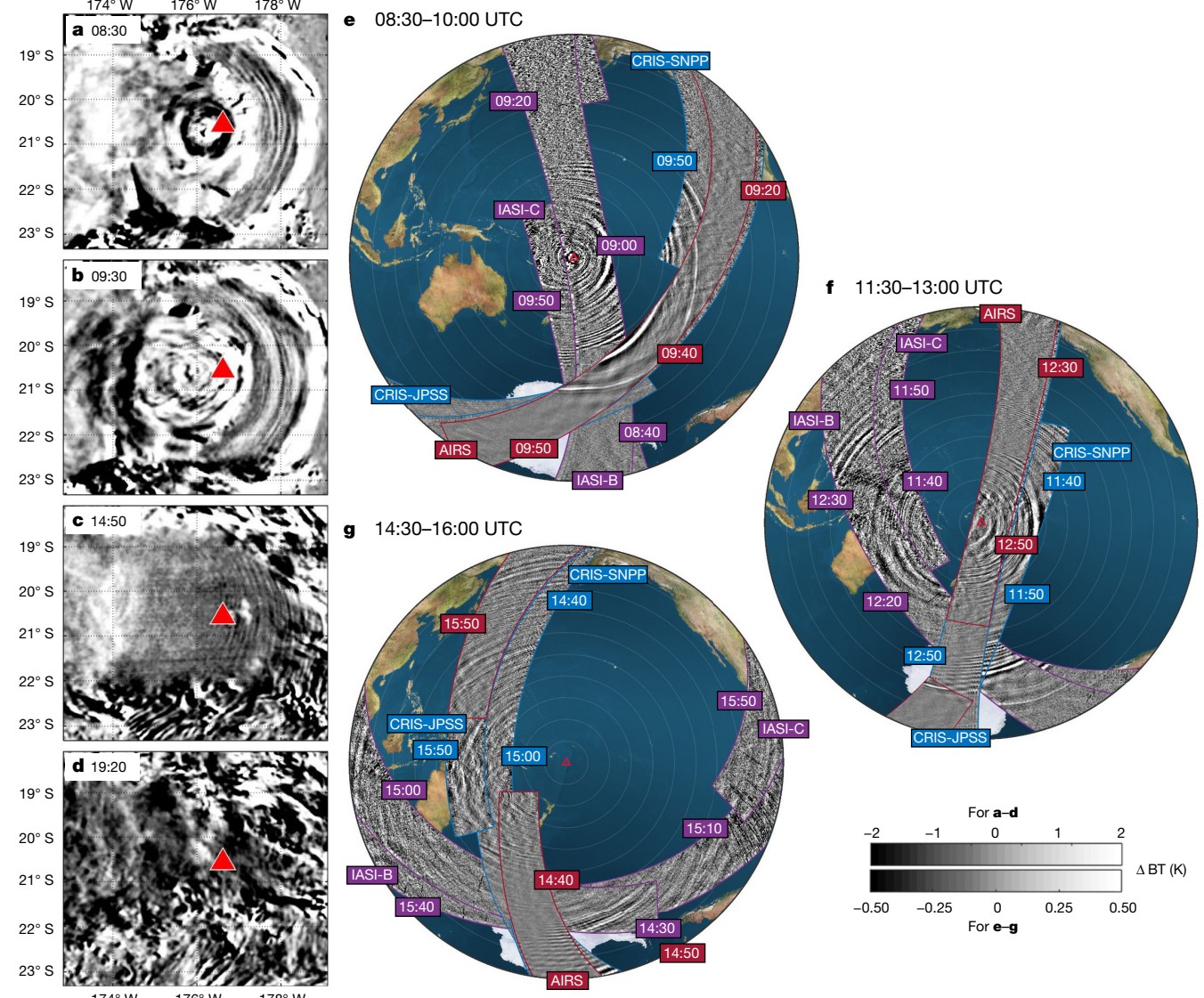

**Fig. 3 | Post-eruption gravity wave activity. a–g**, Activity in and around the volcanic plume as observed by GOES (**a**–**d**) and over the entire Pacific basin as observed by AIRS, CrIS and IASI (**e**–**g**). For **e**–**g**, coloured labels indicate individual satellite overpass times for context, with AIRS labelled in red, CrIS in blue and IASI in purple. Note that the colour scales in **a** and **b** saturate significantly and values extend to ±8 K.

observed spectrum at a given location arises has been complicated by fundamental problems in distinguishing the source of a wave from the pathway it has taken to the observation[29]. Being able to separate these problems would lead to major advances in simulating and parameterizing gravity waves in next-generation weather and climate models. The Hunga Tonga eruption represents an important natural experiment in this area: the volcano was a clearly identifiable near-point source, produced gravity waves across a broad range of spatiotemporal and frequency scales, and these waves were observed by a diverse array of instruments worldwide.

Although the greater than 150 m s$^{-1}$ phase speed waves produced by the initial eruption are unusual at heights below the mesosphere, models in current use do routinely parameterize gravity waves with phase speeds as high as 100 m s$^{-1}$ at altitudes as low as 16 km (ref. [42]), similar to a large fraction of those we observe after the main eruption. In addition, waves in the poorly instrumented mesosphere and above can routinely have speeds of hundreds of metres per second[43], and observations of what is an extreme case in the better-instrumented stratosphere could provide useful insights for future research in this area.

As such, simulating this eruption in atmospheric models, whether as a point convective source or in a dedicated volcanic simulation, could provide major insight into the strengths and deficiencies of models operating across all levels of the atmospheric system. Although current-generation global-scale weather models cannot reproduce these waves because of their relatively limited spatial and temporal resolution and the Courant–Friedrichs–Lewy condition, the waves can be directly resolved by large eddy simulations[44] and similar specialist models[43], albeit only currently for relatively small geographic regions. For such models, the wave observations documented here, made possible only by the exceptional strength of the event, provides a rich source of data to simulate, parameterize and understand these wave types, all of which will be of high relevance to weather and climate models.

Finally, we note that the observed propagation of these waves can also be used as a test of how well models reproduce the bulk atmosphere, by comparing propagation delays for the observed Lamb and gravity waves with those reproduced by simulated waves passing through the model atmosphere. These could provide important information quantifying how well current and future models represent atmospheric

winds, temperatures and density structures, particularly if constrained to the initial conditions of 15 January 2022.

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

## Methods

### Explosive energy estimate from surface pressure data

We estimate the explosive energy associated with the eruption using three separate approaches. All three give a value in the range of 10–28 EJ.

(1) Waveform based on a nuclear explosion: it was suggested in ref. [45] that the energy yield of an explosion in the atmosphere can be calculated as $E = 13p \sqrt{[r_e \sin(r/r_e)]} H_s (cT)^{3/2}$, where $p$ is the measured pressure anomaly, $r$ the distance from the explosion, $r_e$ the Earth's radius, $H_s$ the atmospheric scale height, $c$ the speed of the wave and $T$ the time separation between the first and second peaks of the pressure disturbance. From available pressure-station data at distances ranging from 2,500 to 17,500 km from Hunga Tonga (Extended Data Fig. 1b), this provides an estimate of around $20 \pm 8$ EJ.

(2) Waveform based on previous volcanic eruptions: it was estimated in ref. [46] that the explosive energy of a volcanic eruption is $E = \frac{2\pi H_s \sin(\theta)}{\rho c} \int_{t1}^{t2} p^2 \, dt$, where $\theta$ is the distance from the eruption in degrees, $\rho$ the Earth's surface air density, $t$ is time and $t1$ and $t2$ are the start and end times of the anomaly (different for each station). This gives an estimate of around 10 EJ.

(3) Estimated pressure force: assuming the pressure anomaly spreads under an even cloud of area $A$, then the work done by the pressure impulse over a column of height $h_c$ is $W = pAh_c$. For an area of radius 200 km and a pressure change of 5 hPa, this gives a work estimate around 18 EJ.

### Estimate of Lamb wave phase speed

We use the approach given in ref. [27] and initial-release data from the European Centre for Medium-Range Weather Forecasts' Fifth-Generation Reanalysis (ERA5T) to calculate the expected speed of the Lamb wave. We first compute the local speed of sound as $c_s(z) = k\sqrt{T}$, where $z$ is the altitude, $T$ the local temperature and $k = 20.05$ m s$^{-1}$K$^{-1/2}$. For a Lamb wave, where energy density decays exponentially with height, energy density is $E(z) = C \exp(-z/H)$ where $C$ is a constant term that subsequently cancels in our calculation, and $H$ is

$$H = \frac{(c_s^2)}{(2 - \gamma)} g,$$

for a ratio of specific heats $\gamma$, which we set to 1.4, and acceleration due to gravity $g$, which we set to 9.80665 m s$^{-1}$. We then calculate the phase speed of the Lamb wave as a vertical mean of the speed of sound weighted by energy density, that is,

$$c_m^2 = \frac{\int_0^\infty [c_s(z) + u(z)]^2 E(z) \, dz}{\int_0^\infty E(z) \, dz},$$

where $u$ is the local wind speed.

For ERA5T meteorological output for 15 January 2022 at the 04:00 UTC timestep, this gives a phase speed of 313–318 m s$^{-1}$. Similar results are obtained using the 05:00 UTC timestep. Our calculation omits the contribution of altitudes above 80 km to the energy density calculation as ERA5 data do not extend above this level, but as energy density decreases exponentially with height this contribution should be small.

### Gravity wave speed limit calculation

Linear wave solutions to the Navier–Stokes equations of the form $A \exp[i(kx + mz - \hat{\omega}t)]$ satisfy the dispersion relation [22] of ref. [7], which is fourth-order in intrinsic frequency $\hat{\omega}$. For higher-frequency waves where $f^2 \ll \hat{\omega}^2$ and simplifying to planar two-dimensional propagation, that is, $l = 0$, we can rewrite this as a fourth-order equation in intrinsic phase speed $\hat{c} = \hat{\omega}/k$,

$$\frac{\hat{c}^4}{c_s^2} - \hat{c}^2 \left(1 + \frac{1}{4H^2k^2} + \frac{m^2}{k^2}\right) + \frac{N^2}{k^2} = 0.$$

Letting $x = \hat{c}^2$ gives a quadratic form of the equation

$$ax^2 + bx + c = 0$$

where $a = 1/c_s^2$, $b = -(1 + 1/(4H^2k^2) + m^2/k^2)$ and $c = N^2/k^2$, with solution

$$\hat{c}^2 = \frac{-b \pm \sqrt{b^2 - 4ac}}{2a}.$$

The positive root describes acoustic wave solutions and the negative root internal gravity waves. Allowing the vertical wavenumber $m \to 0$ gives the curve $\hat{c}_{max}(k)$, the maximum phase speed for gravity waves before total internal reflection would prevent their vertical propagation. This limit is

$$\hat{c}_{max}^2 = \frac{c_s^2}{2}[1 + (4H^2k^2)^{-1} - \sqrt{[1 + 1/(4H^2k^2)]^2 - 4N^2/(c_s^2k^2)}\,]$$

and is shown as a function of horizontal wavelength $k^{-1}$ in Extended Data Fig. 6. Our results for the wave properties produced by Hunga Tonga are consistent with previous theoretical work considering normalized full spectra of acoustic and gravity waves[47,48].

### Airglow imagery processing

Airglow data have been obtained from the all-night cloud cameras at the Gemini Observatory on Mauna Kea, Hawaii. This assumed height layer is based on the colour of the airglow and spectral range of the cameras used at Gemini, which are both consistent with the hydroxyl (OH) airglow layer. There are five such cameras, one of which is aimed at a near-vertical angle (with a slight offset determined from study of the star field), and we use this image to identify the arrival time of the first wave packet using the image time stamp—this time is 08:48:53 UTC. At a distance of 4,964 km and using an explosion time of 04:28:48 UTC, this gives a phase speed of 318.12 m s$^{-1}$. Further analysis using the other four cameras from the Gemini observatory gives results consistent with this.

### AIRS, CRIS and IASI

We use brightness temperature observations associated with radiances in the 4.3 μm and 15 μm carbon dioxide absorption bands of AIRS, CrIS, IASI-B and IASI-C[49] on 15 January 2022. These instruments can directly resolve stratospheric waves with vertical wavelengths of more than around 15 km and horizontal wavelengths of more than around 30 km, and typically provide twice-daily near-global coverage for each instrument in near-real time with an orbit approximately every 90 min. Perturbation fields suitable for spectrally and visually analysing wave signatures are produced by subtracting a fourth-order polynomial in the across-track direction from the data, consistent with previous work using these data[6,50].

### CIPS

Imagery from the nadir-viewing CIPS instrument is analysed for the presence of deviations from a smooth model background of Rayleigh scattered ultraviolet sunlight (265 nm). The model removes the geometrical dependence of the observation and large-scale geophysical variability of the observed albedo. The data are binned to a uniform $7.5 \times 7.5$ km$^2$ grid, allowing for observations down to a horizontal wavelength of 15 km. The altitude kernel limits sensitivity to vertical wavelengths of more than around 10 km, with a mean altitude of the contribution at at altitude of approximately 55 km. The satellite is in a sun synchronous polar orbit with an equator crossing currently near noon.

## GOES/Meteosat-SEVIRI

We use data from band 13 of GOES-EAST and GOES-WEST, and band 5 of Meteosat-SEVIRI. These instruments image the Earth's disc at a spatial resolution of 3 km (at nadir) and a temporal resolution of 10 min (15 min for SEVIRI). Raw radiance data have been converted to brightness temperatures based on the centre wavelength of the channel filters and then differenced between adjacent timesteps to highlight wave structure.

## TEC

TEC observations were derived from dual-frequency GPS receivers in the New Zealand GeoNet and the NOAA CORS Networks. Satellite to ground GPS signals were processed following the method of described in ref. [51], and the detrended total electron content (dTEC) values are projected onto an ionospheric shell altitude of 250 km, chosen to be near the F-layer peak height[52]. The dTEC are then analysed to investigate the TID parameters. The data are binned onto a 1 min × 5 km time–distance grid; this suppresses peak values, but improves the visual clarity of the figures. All quoted TEC values are taken from these binned data and thus slightly underestimate TEC magnitudes.

## Data availability

Airglow data are available from https://www.gemini.edu/sciops/telescopes-and-sites/weather/mauna-kea/cloud-cam/allnightlong.html. They were obtained under a Creative Commons Attribution 4.0 International License issued by the NSF's NoirLab. AIRS and CrIS data are available from the NASA Goddard Earth Sciences Data and Information Services Center, https://disc.gsfc.nasa.gov/. CIPS data are available from the Laboratory for Atmospheric and Space Physics at the University of Colorado Boulder, https://lasp.colorado.edu/aim/. ERA5 data are available from the Climate Data Store, https://cds.climate.copernicus.eu. GOES data are available from the NOAA Geostationary Satellite Server, https://www.goes.noaa.gov/. IASI data are available from the IASI Portal, https://iasi.aeris-data.fr/. Meteosat-SEVIRI data are available from the EUMETSAT Data Portal, https://navigator.eumetsat.int/product/EO:EUM:DAT:MSG:HRSEVIRI. Surface pressure data are archived in a Zenodo repository, https://doi.org/10.5281/zenodo.6575810. Of the 36 pressure time series used in this study, 19 are directly included in this repository, including that shown for Tonga in Extended Data Figs. 3 and 8. The repository also includes a table of phase speed estimates calculated for use in Fig. 2 and Extended Data Fig. 1. A further 11 time series used to compute values in the table were obtained from existing public repositories, and the data description of the Zenodo repository specifies their locations. Six time series from the Australian Bureau of Meteorology could not be archived owing to licensing terms; from these, we have included derived estimates of phase speed in the table, and the raw data can be obtained for a fee from the Bureau of Meteorology from http://www.bom.gov.au/climate/data-services/. TEC data are available from https://www.geonet.org.nz/ and https://geodesy.noaa.gov/CORS/.

## Code availability

All software used is either already publicly available, implements equations provided in the Methods section directly or only plots data.

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

**Acknowledgements** C.J.W. is supported by a Royal Society University Research Fellowship, reference no. UF160545. C.J.W. and N.P.H. are supported by NERC grant no. NE/S00985X/1. M.J.A. and C.E.R. were supported by a NASA Heliophysics DRIVE Science Center (grant no. 80NSSC20K0628). C.N.M. was supported by NERC Fellowship NE/P006450/1 for work underpinning this research. C.C. and M.B. received funding from the European Research Council (ERC) under the European Union's Horizon 2020 and innovation programme (grant agreement no. 742909, IASI-FT advanced ERC grant). J.C. was supported by the NASA AIM Small Explorer Program, contract no. NAS5-03132. The Australian Institute of Marine Sciences, the Australian Bureau of Meteorology and the Tongan Met Office are thanked for provision of surface station pressure data. We thank I. Krisch, N. Kaifler and B. Kaifler (all at the DLR, Oberpfaffenhofen, Germany) for assistance with preliminary data analysis, A. Boynard (LATMOS, Paris, France) for providing the H$_2$O IASI data, S. Proud (RAL) for correcting some details of the geostational imager measurements and E. Gryspeerdt (Imperial College, London, UK) for independent confirmation of the Lamb wave trigger time.

**Author contributions** C.J.W. administered the study. C.J.W., L.H. and S.M.O. conceptualized the study. L.H., M.B., J.C., C.C., C.N.M. and C.E.R. performed data curation. C.J.W., N.P.H., M.J.A., M.B., C.N.M., F.P. and L.H. performed the formal analysis. C.J.W. and C.C. were responsible for funding acquisition. C.J.W., N.P.H., M.J.A, M.B., C.N.M., F.P. and L.H. were responsible for the methodology. C.J.W., N.P.H., M.J.A, M.B., C.N.M., F.P. and L.H. wrote the software. C.J.W., N.P.H., M.J.A., M.B., F.P, J.C. and C.C. were responsible for the visualization. C.J.W., N.P.H., M.J.A, L.H, C.N.M., F.P., J.C. and S.M.O. wrote the original draft of the manuscript. All authors performed the investigation and reviewed and edited the manuscript.

**Competing interests** The authors declare no competing interests.

**Additional information**
**Correspondence and requests for materials** should be addressed to Corwin J. Wright.

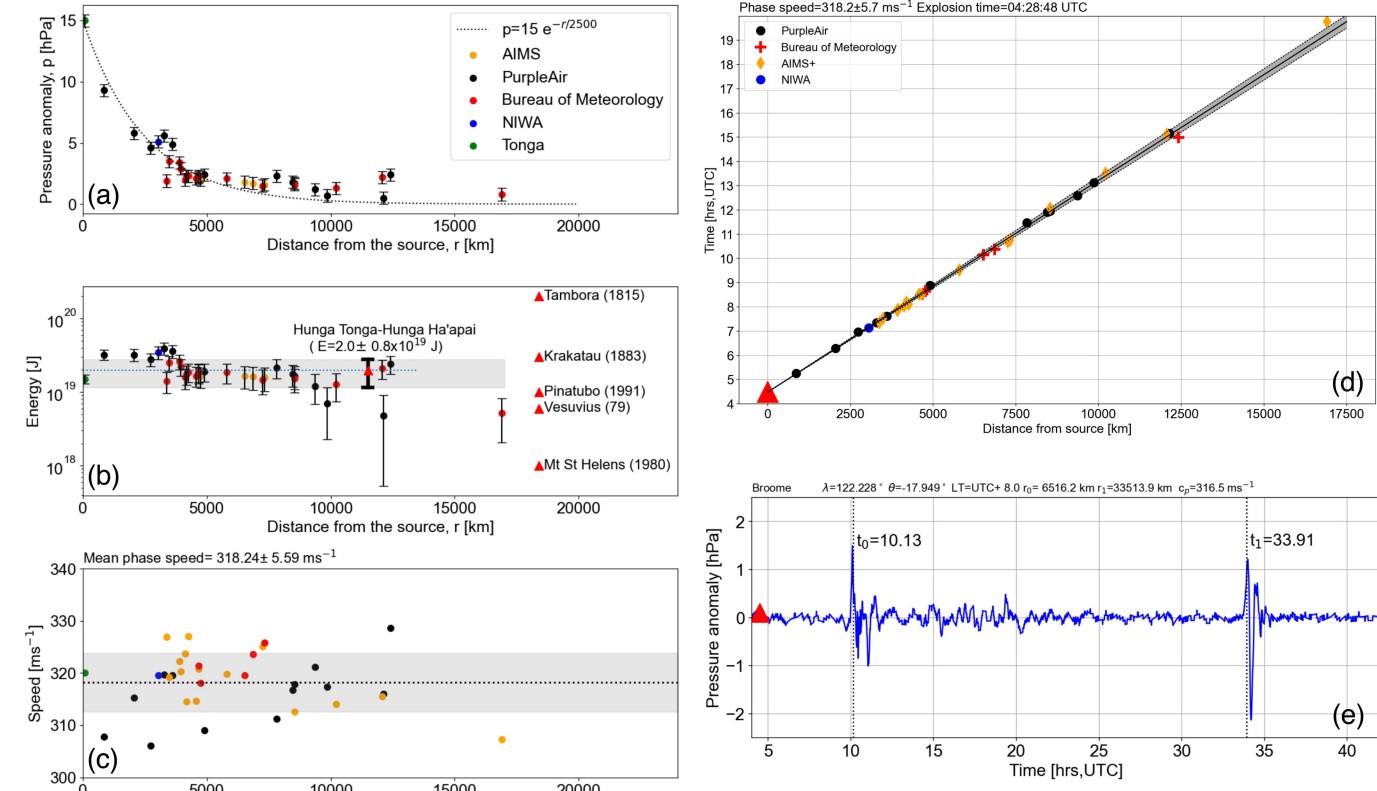

**Extended Data Fig. 1 | Eruptive energy and Lamb wave speed derived from surface pressure changes. a–d**, Estimates of (**a**) Lamb-wave-induced pressure anomaly, (**b**) eruption explosive energy, (**c**) Lamb wave phase speed and (**d**) time of primary explosion, as computed from surface pressure data. **e**, Time series of measured pressure anomaly at Broome, Australia. Data in all cases are derived from surface pressure stations, with the exception of reference values for other eruptions which are derived from ref. [2]. Error bars on panels **a**, **b** are conservatively set to 0.5 hPa.

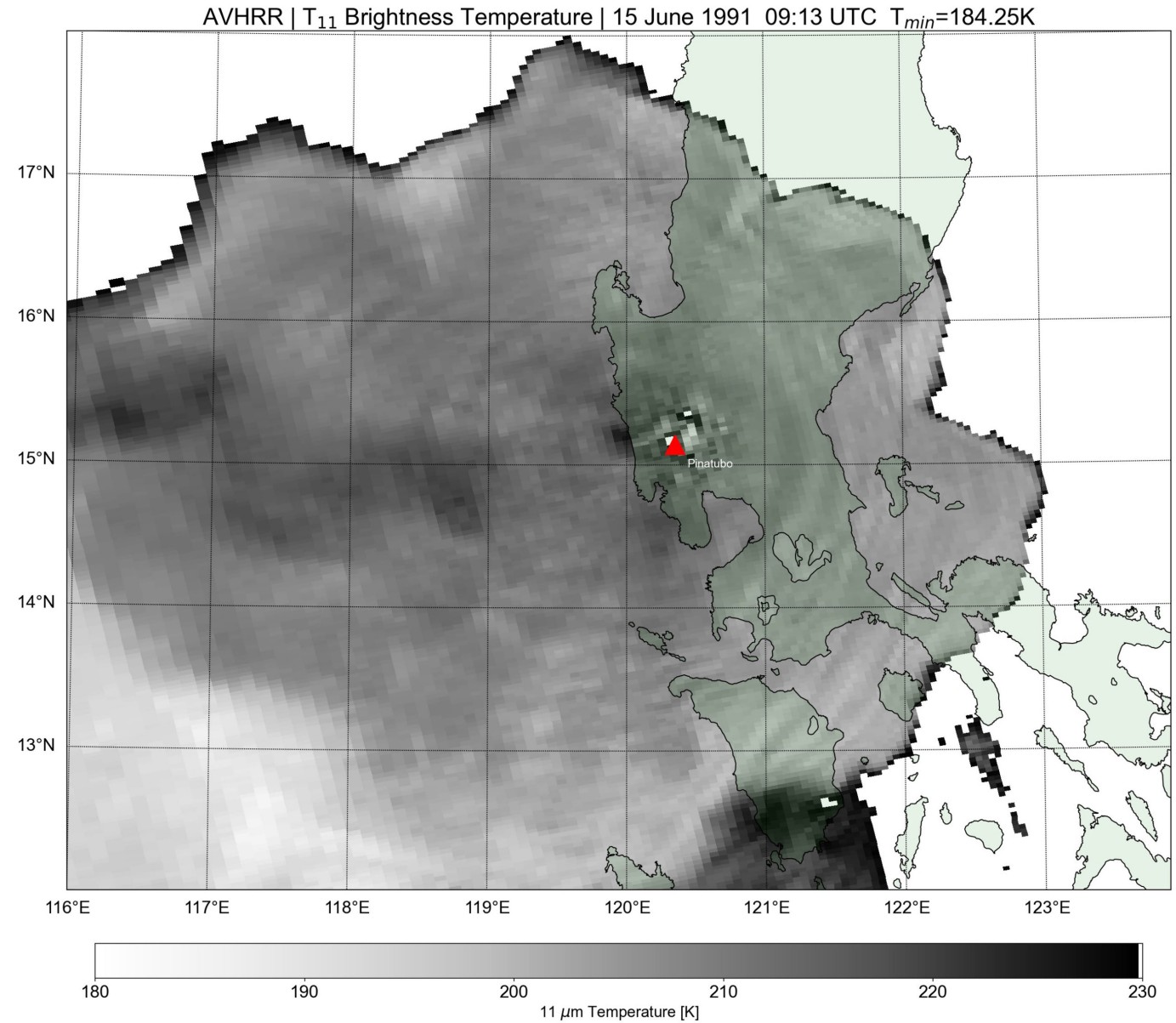

**AVHRR | T$_{11}$ Brightness Temperature | 15 June 1991  09:13 UTC  T$_{min}$=184.25K**

11 $\mu$m Temperature [K]

**Extended Data Fig. 2 | Reprocessed data for the 1991 Pinatubo eruption show evidence of gravity wave activity in the eruptive plume.** Brightness temperature measurements over the 1991 Pinatubo eruption plume, as observed by the Advanced Very High Resolution Radiometer. Phase fronts can be seen faintly in the cloud radiating from a point slightly west of Pinatubo.

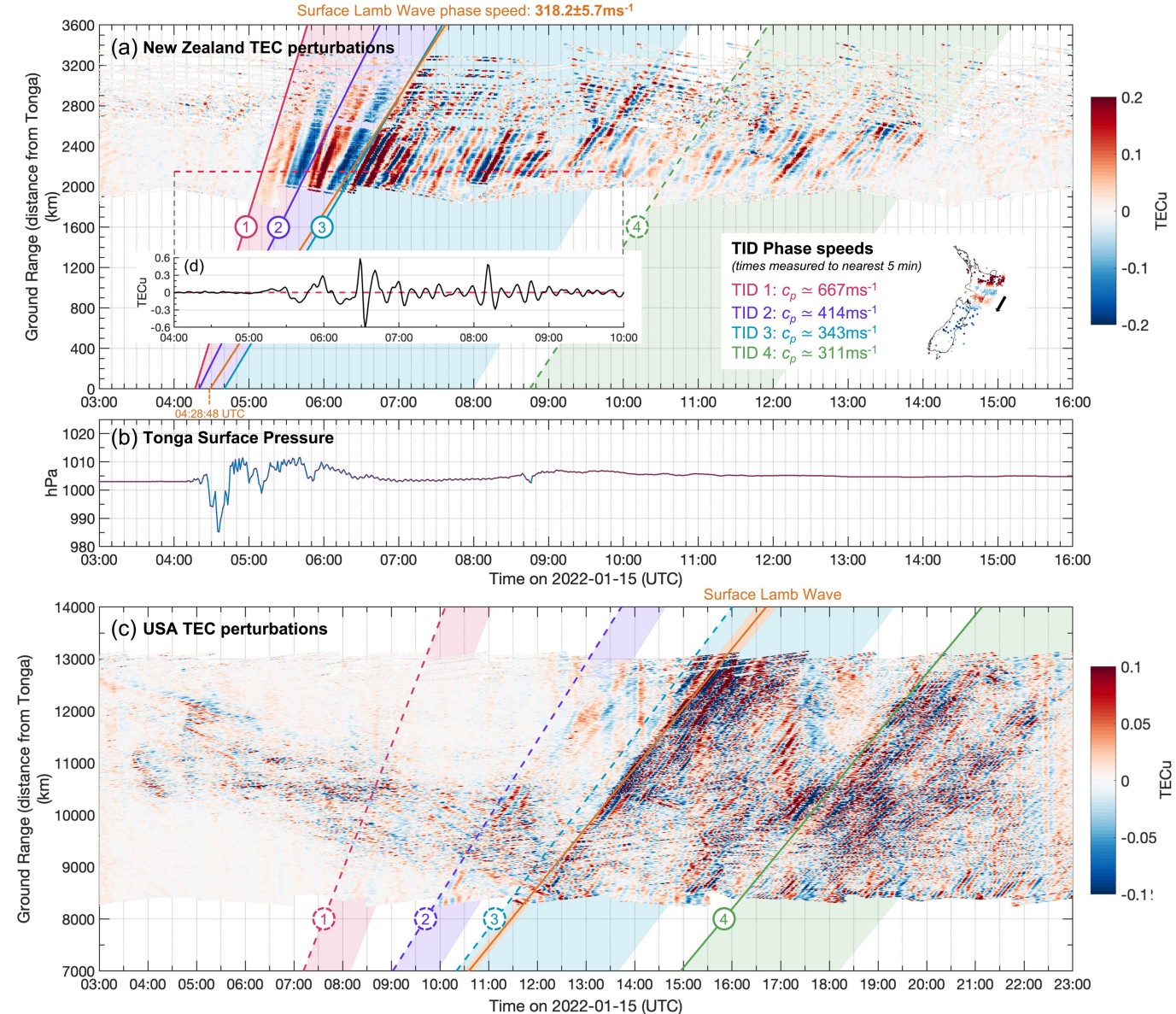

**Extended Data Fig. 3 | Evidence of waves in the ionosphere over New Zealand and North America triggered by the Hunga Tonga eruption.** Time-distance plots of ionospheric disturbances over New Zealand and the United States, computed from GNSS-TEC perturbation data. **a**, TEC perturbations as a function of distance from Hunga Tonga and time over New Zealand. **b**, Surface pressure at Tonga, approximately 60 km from Hunga Tonga. **c**, TEC perturbations as a function of distance and time over North America. **d**, Cross-section through panel **a** for selected period.

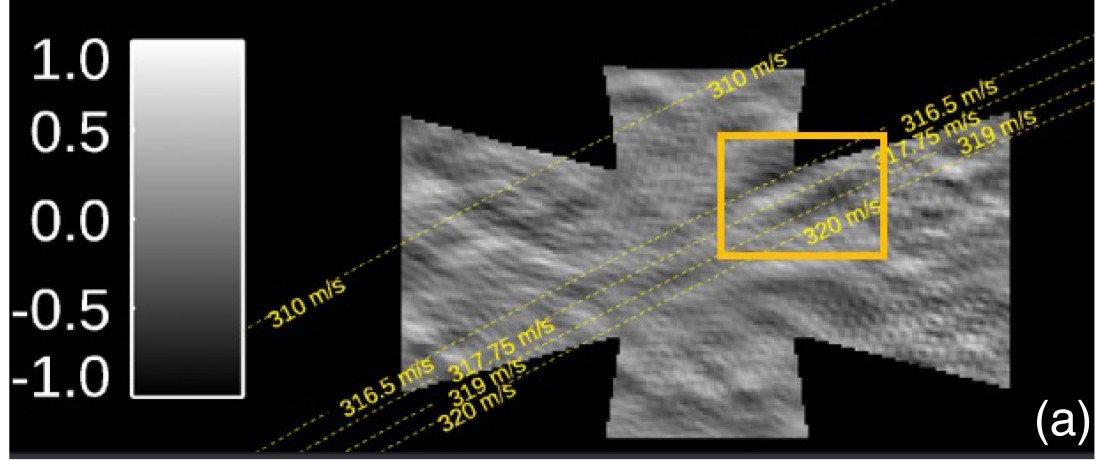

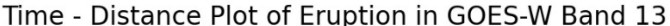

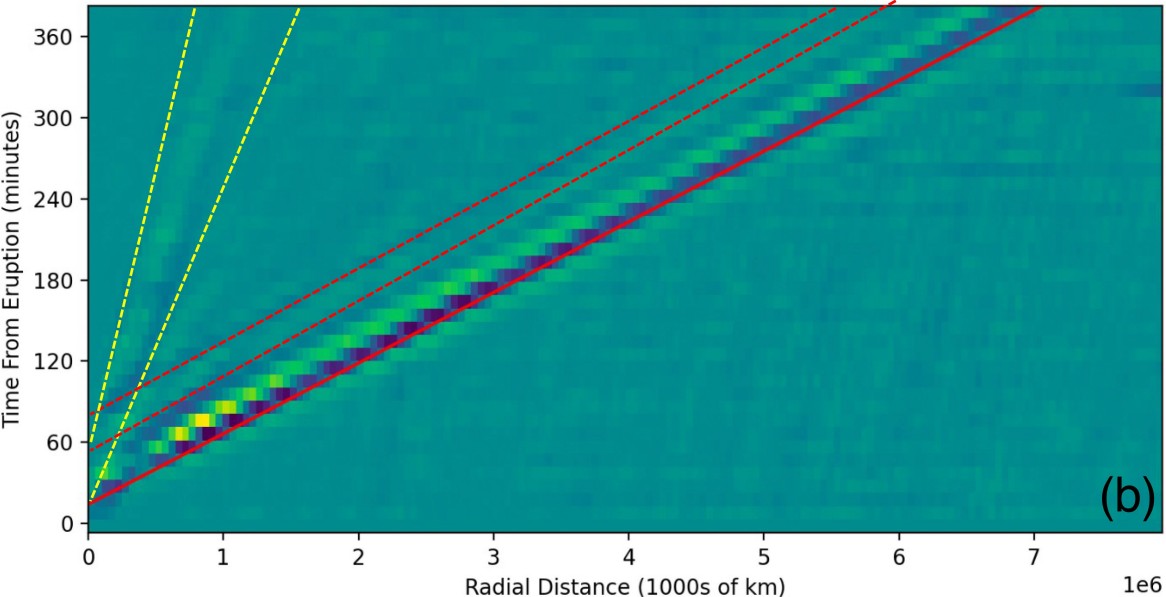

**Extended Data Fig. 4 | The waves generated by the eruption propagated up to the mesosphere and travelled horizontally at speeds consistent with their types. a**, Lamb wave as observed by CIPS (centred at 24°S 309°E, 12 300 km from Hunga Tonga, and recorded 10.75 h after the eruption). In these data, the Lamb wave is extremely close to the instrument noise floor and statistical tests were carried out to confirm that the small signal seen is consistent with the expected speed and wavelength of the Lamb wave. **b**, Time-distance spectrum derived from GOES 10 um channel, with Hunga Tonga located at the origin. Red solid line identifies the primary Lamb wave, red dashed lines identify weaker secondary Lamb waves, and yellow dashed lines outline the limits of the dispersive gravity waves in the initially released packet.

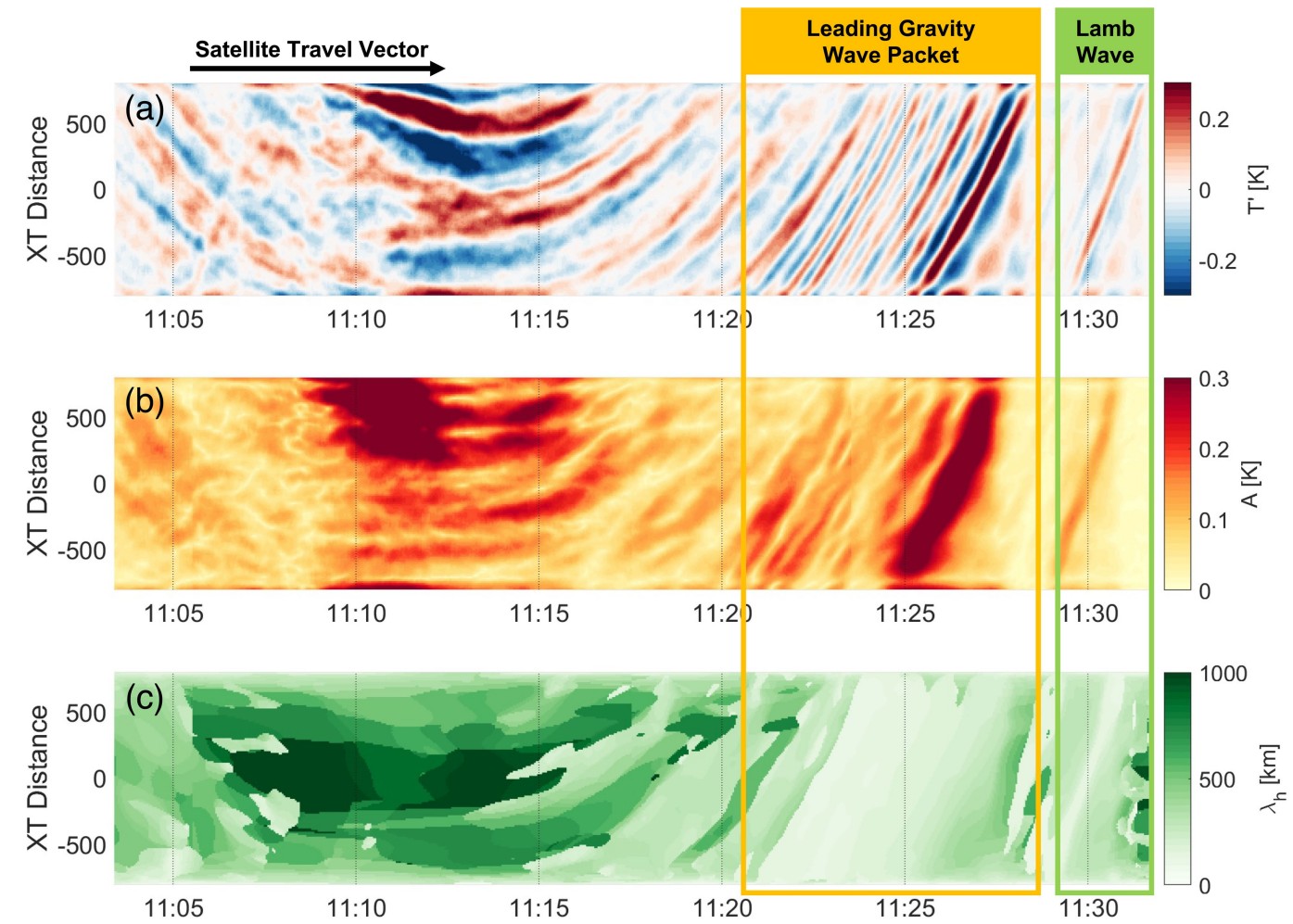

**Extended Data Fig. 5 | Spectral analysis provides quantitative details of stratospheric waves generated by the eruption.** 2D S-Transform[53] (2DST) estimates of gravity wave properties measured by AIRS in a descending-node pass over the Pacific Ocean on the 15th of January 2022. **a**, Temperature perturbations relative to a fourth-order polynomial fit across track. **b**, amplitudes estimated from these perturbations using the 2DST. **c**, Horizontal wavelengths estimated from these perturbations using the 2DST.

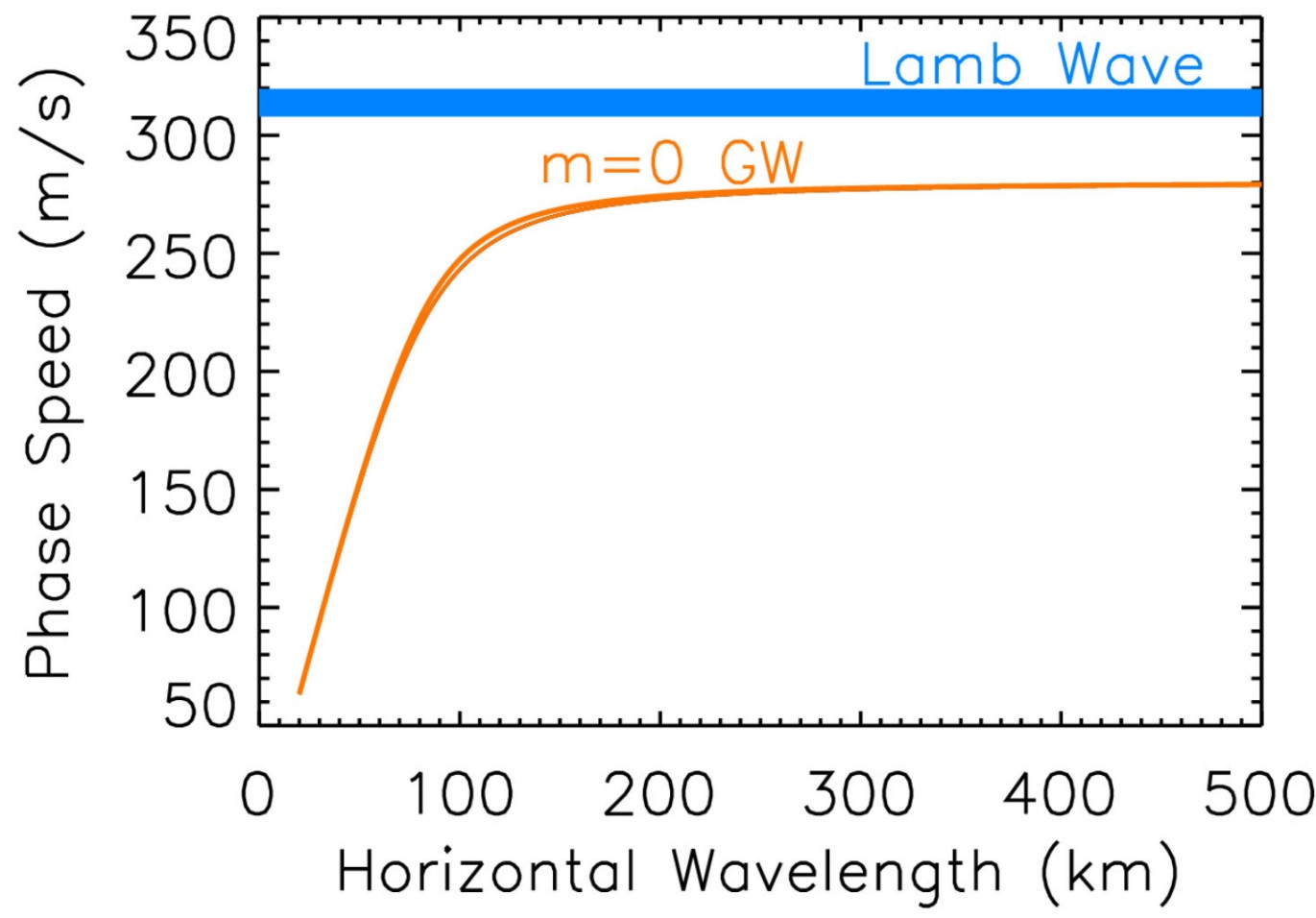

**Extended Data Fig. 6 | The gravity waves generated by the eruption travelled close to their maximum phase speed limit.** Expected maximum speed of a gravity wave packet relative to the observed Lamb wave, as a function of horizontal gravity wave wavelength. Blue line thickness represents the range of Lamb wave propagation speeds that we compute from AIRS, with the fast edge being approximately equal to the speed of the surface pressure signal. Orange lines represent the fast limit of gravity wave phase speeds versus horizontal wavelength, which is in the limit that the vertical wavenumber $\rightarrow 0$. This has been calculated using the upper and lower Lamb wave speeds as the sound speed for this calculation, shown as two closely overlaid orange lines.

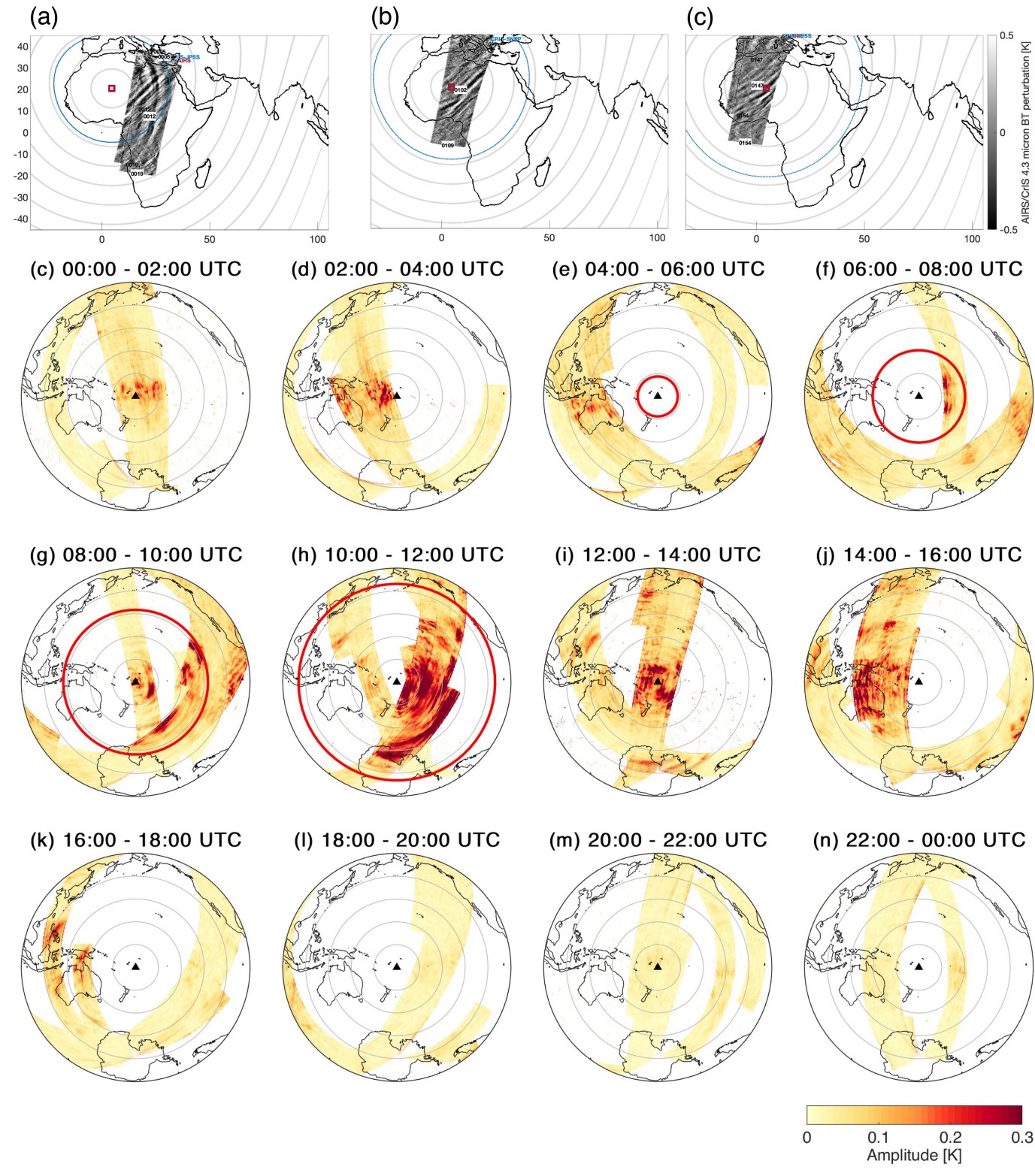

**Extended Data Fig. 7 | Gravity waves produced by the eruption traversed the entire globe and dominated the Pacific basin following the eruption. a–c,** Transit of the leading gravity wave packet over the antipode in CrIS and AIRS 4.3 μm data. **(d–o,** GW amplitudes over Pacific computed from AIRS, IASI and CrIS 4.3 μm data using the 2DST[38].

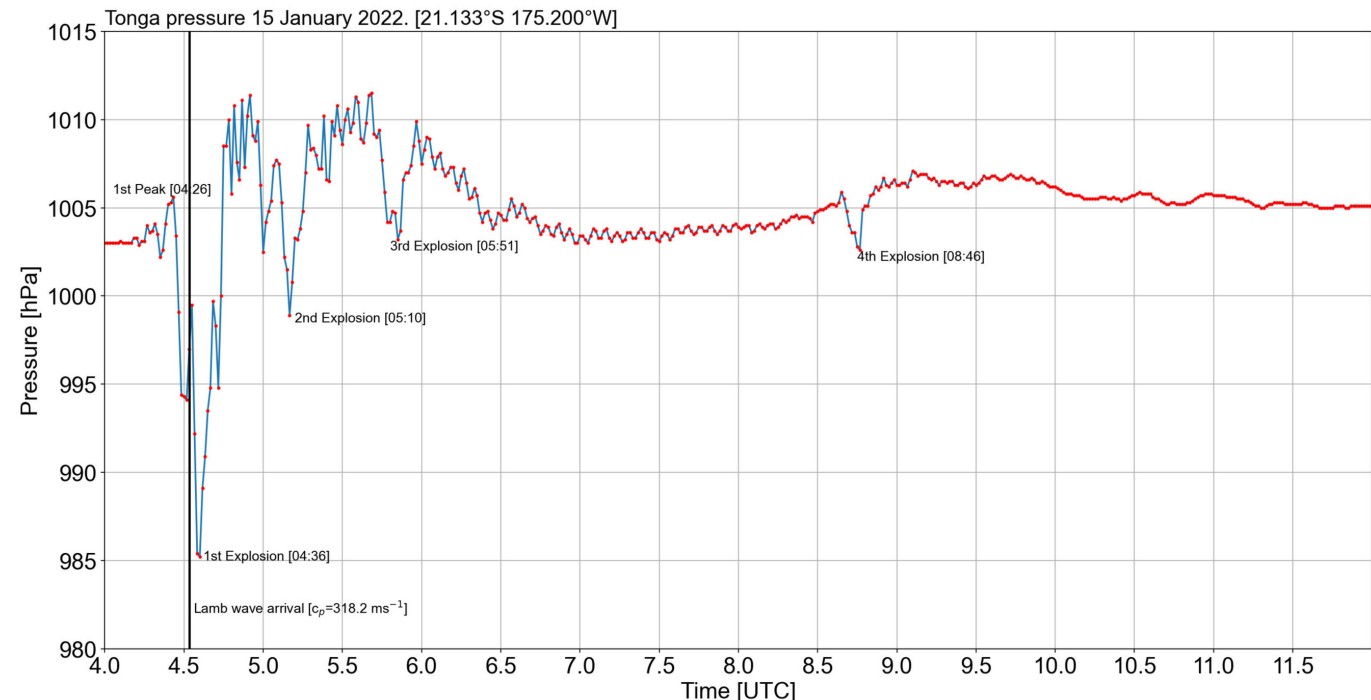

**Extended Data Fig. 8 | Surface pressure data show evidence of multiple subsequent explosions.** Surface pressure station measurements from 04:00–12:00 UTC from Tonga, approximately 64 km from Hunga Tonga. Note the multiple explosions after the initial primary Lamb wave trigger.

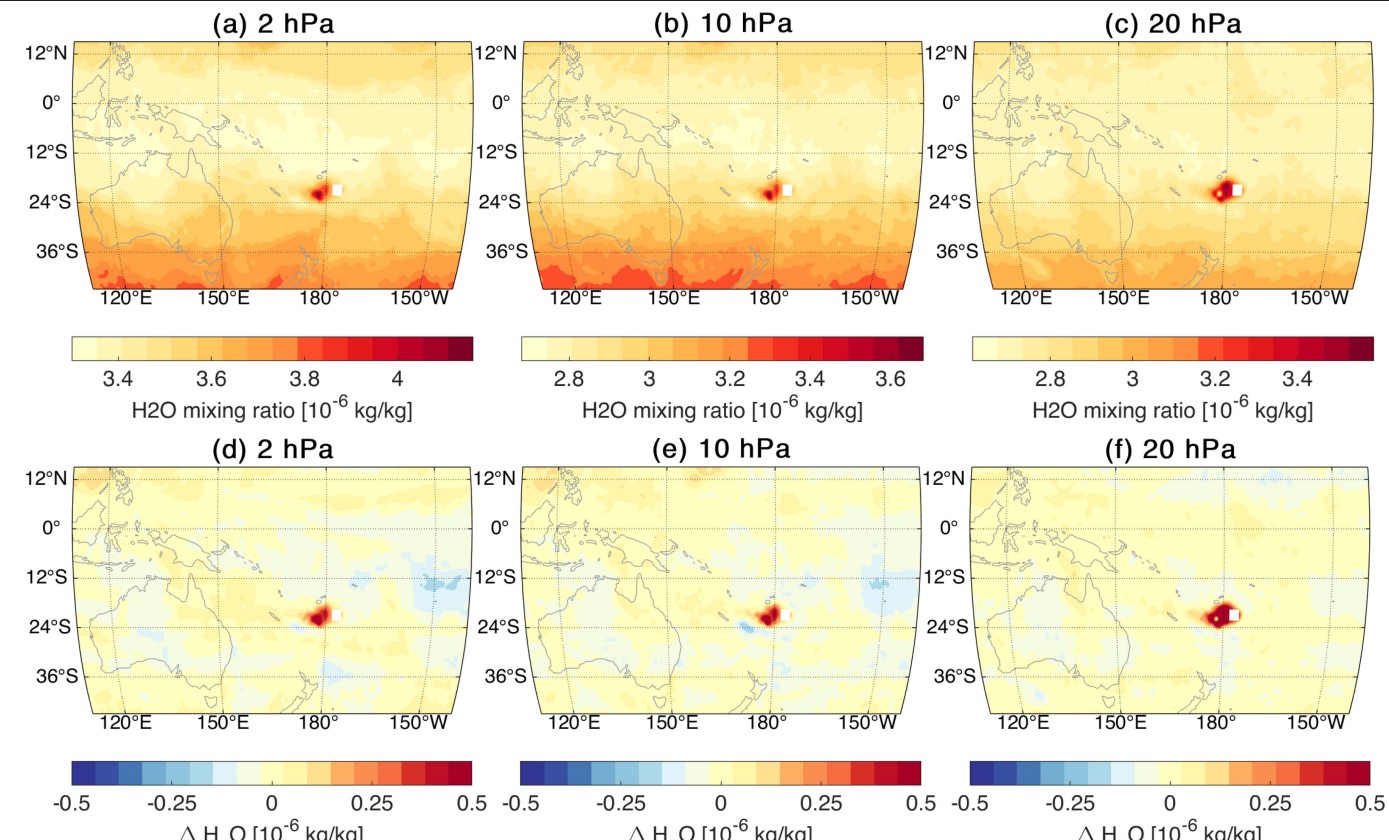

**Extended Data Fig. 9 | Water vapour observations are consistent with our proposed eruptive energy transfer mechanism.** 1x1 degree maps of IASI-B and IASI-C water vapour mixing ratio at the 2, 10 and 20 hPa levels for the 15th of January 2021, using nighttime data. **a–c**, show the data as absolute values and **d–f** as a difference from the local mean for January 2021. White squares indicate a lack of data owing to retrieval failure, most likely due to the highly anomalous atmospheric state associated with the eruption plume.

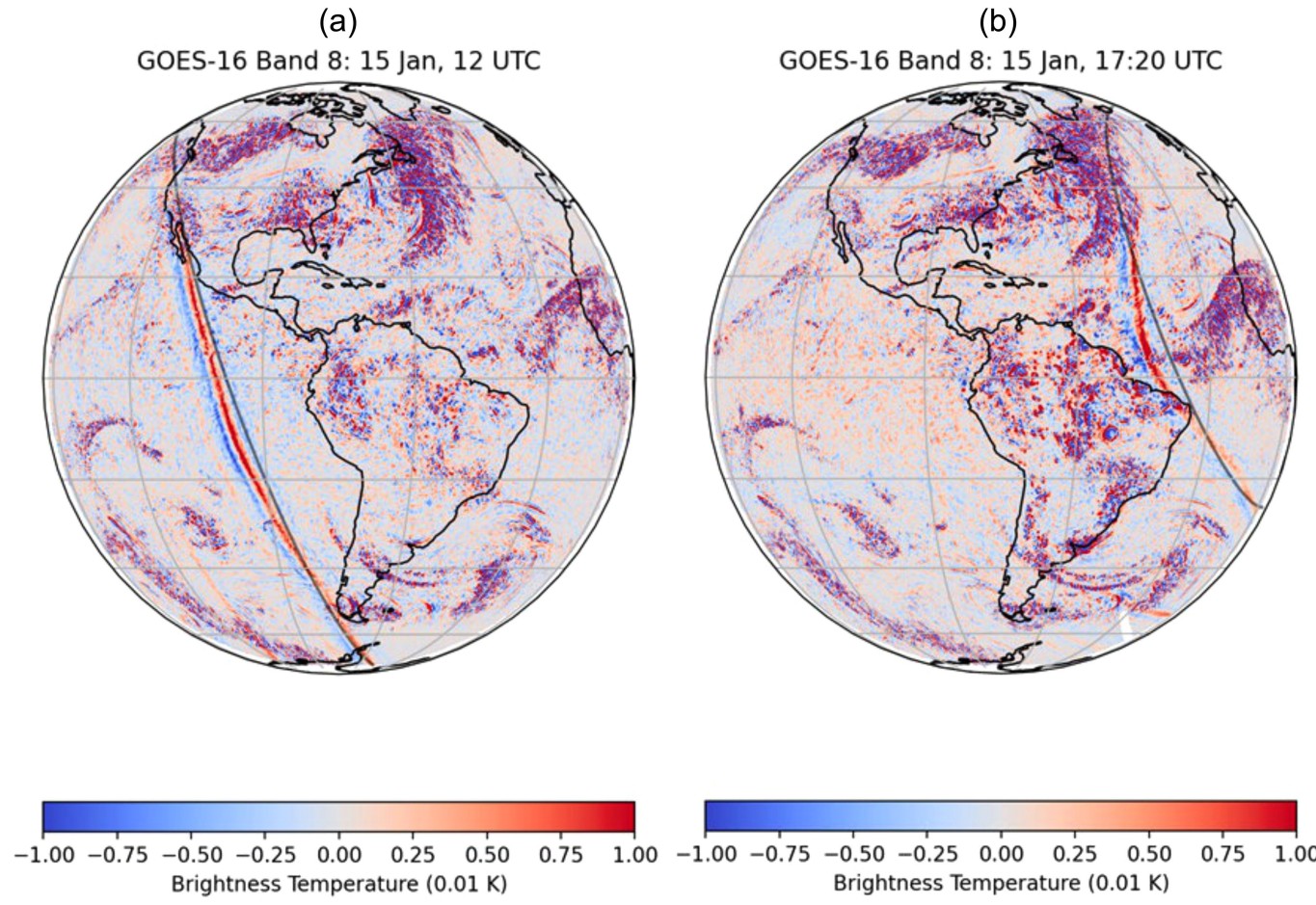

**Extended Data Fig. 10 | The Lamb wave shows evidence of slowing down over South America.** Filtered data from GOES' IR channel showing the Lamb wave (strong blue/red/blue alternating lines) before (left) and after (right) passage over South America. Overlaid grey line shows the the expected location of the phase front assuming uniform progression. An increased deviation from this expected line is seen in the portion of the wave which passed over the northern half of South America.