## [Peer Review File · Nature]

Manuscript Title: Surface-to-space atmospheric waves from Hunga Tonga-Hunga Ha'apai eruption

Reviewer Comments & Author Rebuttals

Reviewer Reports on the Initial Version:

Referees' comments:

Referee #1 (Remarks to the Author):

This is a unique and very well-executed manuscript that shows for the first time how atmospheric waves generated by a single known source can propagate around the entire planet.

The authors are to be commended for producing such a rich and detailed analysis only a matter of weeks after the eruption of Tonga.

The analysis appears to be technically correct and is illustrated by detailed and beautifully produced figures.

My main comments below concern the interpretation and labeling of the gravity waves, their relevance to improving weather and climate models, and the connection of this manuscript with another recently published study (from different authors) that appeared a few days ago.

My disposition is that the manuscript should be published pending consideration of the below comments. It would be a valuable addition to the peer-reviewed literature. Whether the content and scope are wide enough to warrant publication in Nature, as opposed to a more specialist journal, is of course an editorial judgement call.

MAJOR COMMENTS

1. The internal gravity waves studied here are very fast, moving at around 250 m/s. I confess that it is news to me that gravity waves can propagate this quickly! I am more used to speeds that are an order of magnitude slower than this, and I think many atmospheric scientists will be in a similar position, being familiar with the incompressible form of the dispersion equation on line 370, but not the full compressible form included here. The critical role of the speed of sound in setting the propagation speed makes me wonder whether these waves would not in fact better be labeled as "acoustic-gravity waves" rather than pure gravity waves. In any case, I think it should be clarified that these waves are different from what we usually think of as internal gravity waves in the atmosphere. For example, gravity wave drag parameterizations are not attempting to represent these very fast waves (are they?).

2. The claim that these observations will be useful for improving weather and climate models is speculative and would benefit from additional substantiation, if possible. Numerical models typically require the Courant number ($Co = u \cdot dt/dx$) to be less than around 1 for stability, according to the

Courant-Friedrichs-Lewy (CFL) condition. Assuming a grid spacing of 10 km, which is typical for an NWP model, gravity waves propagating at 250 m/s will not be stably resolved unless the time step is shorter than 40 s, which is currently too computationally expensive to be feasible. For this reason, initialization and data assimilation procedures actively filter out such waves. As gravity waves propagating at these very fast speeds are relatively rare, I am unconvinced that our models' ability (or otherwise) to simulate them has much direct relevance to weather and climate forecasting. My disposition would differ if these were "ordinary" gravity waves moving an order of magnitude slower, which unquestionably play a crucial role in weather and climate.

3. A new study published in the last few days analyzes atmospheric pressure anomalies arising from the Tonga eruption (<https://doi.org/10.1002/wea.4170>). I haven't read it in detail, but I wonder how it connects with the manuscript under review. Can the authors confirm that their analysis goes deeper and further?

MINOR COMMENTS

Line 1: Perhaps the title should refer to *atmospheric* waves, unless that would violate a length limit.

Lines 30-32: Are these phase speeds or group speeds?

Line 66: Electrically neutral?

Line 337: What are the time limits on the integral?

Line 349: The constant 20.05 must be dimensional (although its units are not stated in the manuscript). This means its value depends on the units of T and c_s . Best to account for this, by perhaps stating that $c_s(z) = k \sqrt{T}$, where $k = 20.05 \text{ m s}^{-1} \text{ K}^{-1/2}$.

Line 356: What are the altitude limits on the integrals?

Line 365: Waves are normally written $\exp[i(kx + mz - \omega t)]$. Why is it $+\omega t$ here?

Line 365: Typo: there is a spurious "(" before the "i" in the exponential term.

Referee #2 (Remarks to the Author):

The manuscript reports observations of traveling atmospheric waves launched by the January 2022 Tonga volcanic eruption. The observations consist of satellite and ground-based measurements made for the atmospheric layers at various heights, from the surface to the ionosphere. By analyzing the observational data, the authors identify signatures of Lamb waves and internal gravity waves induced by the eruption. These waves are found to propagate globally. The authors also discuss the sources responsible for the atmospheric waves.

The manuscript presents significant and timely results of the unique eruption event. The novelty of the work lies in the combined observational analysis of atmospheric waves at various heights and the finding of the global propagation of the atmospheric waves. The work contributes to the understanding of lithosphere-atmosphere-ionosphere coupling. However, I think the manuscript can be further improved by clarifying some ambiguities and incorporating more analysis/discussion on the result. My comments are listed below.

Major comments:

1. The vertical structure and propagation of the Lamb waves and gravity waves.

The vertical structure and propagation of atmospheric waves generated from volcanic eruptions have been rarely investigated previously, partially due to the lack of observations. For the Tonga eruption, the authors show clear wave signatures observed at various atmospheric heights, which offers a great opportunity to look into the vertical structure and propagation of the waves. If possible, I think the authors should take the full advantage of the observational data and investigate the vertical propagation of the waves. This would significantly improve the novelty of the work. The authors have mentioned the vertical wavelength in the manuscript. What about the propagation time? How much time it takes for each type of the waves to reach various observational heights? Does the vertical propagation speed of the wave varies with height? Can all the Lamb and gravity waves reach the ionosphere (possible atmospheric filtering of waves)?

2. The implications on weather and climate forecasting.

The impact of the result on weather and climate forecasting is not well justified. The Tonga eruption is a rare event, and its atmospheric impacts are extreme. It is unclear how the simulation of this extreme event would provide insights into the strengths and deficiencies of models for weather and climate forecasting, which are nearly always used for non-extreme conditions. The authors should clarify the applicability of the knowledge learnt from this extreme event to typical weather and climate forecasting.

Minor comments:

1. Lines 46 - 47: The unit of energy is EJ or EJ²?
2. Lines 54, "non-acoustic frequencies": what is the range of non-acoustic frequencies? Are the Lamb and gravity waves lower frequency than acoustic waves?
3. Caption of Figure 2: "Airglow inser" → "Airglow insert"?
4. Line 78, "shockwave": where is it and how to determine that it is a shockwave?
5. Lines 137 - 138: Extended Data Figure 3 should be Extended Data Figure 6?
6. Line 138: where are the "Two low-amplitude wavefronts"? Panels are not labeled in Extended Data Figure 5 (no 5b).
7. Line 147, "consistent in speed": please provide explanation. Which two speeds are consistent?
8. Line 151, "6.15am": what does it mean here?

9. Lines 221 - 223: any references?

10. Lines 383-389 and Figure 2 airglow insert: Are the airglow images taken during the night or the day? The text seems to indicate that these are nighttime airglow images, while the figure shows daytime? Can the authors provide the original airglow images?

11. Extended Data Figure 3: There seems a gap between the TIDs at the west coast of USA and south-eastern USA. Why there is little TID in-between? Does this imply that the TIDs in south-eastern USA are induced by sources other than the eruption?

12. Supplementary Figures 1 and 2: it would be better to add timestamps and colorbar.

Referee #3 (Remarks to the Author):

The manuscript presents an interesting compilation and analysis of various datasets that include observations of acoustic-gravity waves (AGWs) generated during the 2022 Tonga eruptions. Sources of AGWs and their excitation mechanisms are proposed and discussed. The advantage of the study is its multi-layer analysis of various measurements and discussion of their similarities and differences. I find this manuscript to provide the most complete (although very compressed due to page limitation) discussion of fluctuations from vertical evolution of AGWs after Tonga eruption up to now. I believe that the manuscript can be a great Nature report which will attract an attention to the problem of coupled processes in Earth's envelopes.

I provide some commentaries below that may help to improve the representation of the results, clearance of the discussion and story coherency. Overall, my main suggestion is to provide more quantitative analysis to support the discussion of the results and findings.

- L161-181, L202 – although this discussion and explanation can be valid, I would point to the fact that the eruption generated a tsunami. This tsunami could be a source of 1) AGWs of a wide range of periods and wavelengths and observed fluctuations in the near-field, 2) AGWs of a wide range of periods and wavelengths and observed fluctuations all over Pacific Ocean and thus 3) various traveling ionospheric disturbances. Among others, for example <https://agupubs.onlinelibrary.wiley.com/doi/10.1029/2020JA028309> demonstrated that tsunami itself generates packet(s) of wide range of AGWs/GWs (see Figure - 1 in the attachments). Although they may be less prominent at far-field in lower atmospheric observations, these waves may potentially be source of detectable fluctuations in near-field, as well as at far-field at ionospheric altitudes.

They also demonstrated that refracted/trapped/deflected ocean waves near the shore may drive AGWs/GW for hours. Similar was demonstrated with TEC observations (Tsugawa et al. 2011). Signals of ~200 m/s looks also comparable with speeds of tsunami propagations (see Figure -2 in the attachments).

Also,

<https://nctr.pmel.noaa.gov/tonga20220115/> and

<https://agupubs.onlinelibrary.wiley.com/doi/10.1029/2022GL098153> recently demonstrated the arrival of the tsunami to the west coast of the US several hours after pressure pulses, in some way

consistent with TIDs over the US.

To summarize, for the sake of completeness, you may consider adding a tsunami because of undersea eruption and thus AGWs in the atmosphere as sources of detected fluctuations.

- L32/L65 (and wherever it is mentioned) – The authors may reference, e.g., Liu et al. 1982 who presented results of the detection of ionospheric fluctuations generated by (potentially) Lamb waves after the 1980 Mt. St. Helens eruption (<https://doi.org/10.1029/JA087iA08p06281> and <https://agupubs.onlinelibrary.wiley.com/doi/abs/10.1029/JA087iA08p06291>), Also, “Ionospheric and atmospheric disturbances around Japan caused by the eruption of Mount Pinatubo on 15 June 1991” by Igarashi et al. 1993.

- For reader, it could be hard to see phases and their continuity to validate reported speeds from Extended Data Figure 3. Could some parts of the figure be zoomed in?

- Figure 2 – I understand what is meant by “initial” gravity waves, but I think it is not needed here, as it may potentially be misleading (otherwise introduce what is meant by “initial” and/or/ subsequent etc.). I would also add panel letters to each observation depicted (i.e., (a) Airglow, Hawaii, (b) Surface Pressure Stations etc.).

- How you define the altitude of GNSS observations of 100-250 km? Could the signals be detected and discerned from the noise at such low altitudes as 100 km? Why then the shell height was set at the top boundary of 250 km?

- L78 – how to understand, from Figure 1, that visible fluctuations are driven by a shockwave? In the same paragraph, it is mentioned Lamb wave. Not clear what is meant by “mixed packet of waves”. What was mixed? How from Figure 2, Extended Figure 1c,d and Supplementary Figure 1 the reader can find “mixed packet of waves with non-dispersive wavelengths and periods”? What periods and what wavelengths are estimated? What is meant by “large” amplitude? To summarize, it would be great to provide additional quantitative discussion here and adjust figures accordingly.

- L86 – which panel of Figure 1 demonstrates 4 wavefronts? Can they be marked? I would also suggest adding here panels (a), (b), (c) to direct a reader to exact panels that are mentioned in the text.

Is it possible to provide more insight on the speeds with GOES data? A slowdown of waves over South America is mentioned, but it is not clear if waves were slowed down over South America or earlier, in the Pacific Ocean (not sure I can see it from Extended Data Figure 4,b). I think time-distance diagram could be useful to support this discussion.

- L98 – 308 ± 5 and 319 ± 4 m/s – are these 2 numbers for 2 different locations or 2 different observations?

- L491 – 12300 km

- L102 – how did you estimate phase speed of 318 m/s for hydroxyl airglow over Hawaii?

- L103 – what is meant by “uniform” phase fronts? Uniform in what extent? I am not sure that any observations demonstrate vertical structure of fluctuation.

- L107 – Does any of demonstrated instrument or dataset have a sufficient precision to claim that Lamb wave propagated with 319 and 316 m/s in different directions? This is not following from the earlier discussion, where the speeds are suggested with an uncertainty of +/-6 m/s (L81) +/-3 m/s (L85), +/-5 m/s and +/-4 m/s (L98). What are the sampling rates of the data demonstrated?

- L151 – 6.15am

- L151 (see Figure - 3 in the attachments)

Does it mean that TIDs (2) arrived to US later than expected? I am looking at those fluctuations between lines (2) and (3) and distance range ~9000-11000 km. If those TIDs arrived later than (2) in New Zealand – are they driven by the same “source” or different? If different, why they are both named as (2)? How TID(4) consistent with eruption? Again, zooming of regions of discussion could be helpful.

Just an idea for the figure - see Figure - 4 in the attachments (ignore if you don't like it). Meaning to put a pressure plot to the bottom of Extended Figure 3. From first try, the TIDs 4 is kind of consistent with 4th explosion, but not fully. By saying “TIDs 4 are consistent with the wave activity generated over Hunga Tonga in the hours after the primary eruption” – do you mean that TIDs 4 is from Lamb waves generated during explosion 4? I think some clarifications can be provided here.

Figure - 1

Figure - 2

Figure - 3

Figure - 4

Response to Reviewer

Reviewer 1

B.1 Major Comments

1. *The internal gravity waves studied here are very fast ... I confess that it is news to me that gravity waves can propagate this quickly! I am more used to speeds that are an order of magnitude slower than this... being familiar with the incompressible form of the dispersion equation on line 370, but not the full compressible form included here. The critical role of the speed of sound in setting the propagation speed makes me wonder whether these waves would not in fact better be labeled as “acoustic-gravity waves” rather than pure gravity waves. In any case, I think it should be clarified that these waves are different from what we usually think of as internal gravity waves in the atmosphere. For example, gravity wave drag parameterizations are not attempting to represent these very fast waves (are they?).*

The gravity waves that follow the Lamb wave are indeed internal gravity waves - they are not intrinsically different from more commonly observed slower internal gravity waves except for their unusually deep vertical structure, which results from the unusually deep, strong source mechanism. Nevertheless, as the reviewer states, model parameterisations do not usually attempt to represent unusual internal waves of this type (although they do include waves of order 100 m/s, and our modified manuscript now states this explicitly), and we address this in our next response.

2. *The claim that these observations will be useful for improving weather and climate models is speculative and would benefit from additional substantiation, if possible. Numerical models typically require the Courant number ($Co=u*dt/dx$) to be less than around 1 for stability, a*

c
-While the waves produced by the initial eruption are indeed extreme, models such as WACC M do

routinely parameterise waves with phase speeds 100 m/s in the lower stratosphere (Richter et al, 2010, doi:10.1175/2009JAS3112.1), which are comparable to those produced by the convective activity following the initial eruption. In addition, a large fraction of gravity waves in the mesosphere and lower thermosphere do have phase speeds this large or larger, and study of these waves in their relatively well-observed middle atmosphere could provide insight into their underlying physics.

These points were not made in the original draft, and the reviewers are correct to flag this up. To

clarify our meaning we have now modified the text to make these statements explicitly, approximately doubling the length of this section of the text.

3. A new study published in the last few days... doi:10.1002/wea.4170 ... I wonder how it connects with the manuscript under review. Can the authors confirm that their analysis goes deeper and further?

Yes: our analysis goes significantly deeper and further than Harrison (2022). The Harrison study focuses on measured pressure anomalies at ground level in the UK only. This provides local information on the propagation of the Lamb wave over the UK, with a particular focus on a specific station in Reading. Our study is at the global scale, considers the whole depth of the atmosphere, and studies gravity waves generated across the entire day as well as the initial Lamb wave. The Harrison study does provide useful additional evidence of repeated Lamb wave passages around the world which we do not show but do mention, and accordingly we have added it to our reference list.

B.2 Minor Comments

1. **Line 1: Perhaps the title should refer to *atmospheric* waves, unless that would violate a length limit.**

The editor has suggested a new title which incorporates this suggestion.

2. **Lines 30-32: Are these phase speeds or group speeds?**

They are phase speeds, as clarified later in the paper. This has been made clearer in the modified abstract.

3. **Line 66: Electrically neutral?**

Yes: this has been clarified.

4. **Line 337: What are the time limits on the integral?**

Bounds have been added to the integral to clarify that this is over the duration of the observed pressure anomaly.

5. **Line 349: The constant 20.05 must be dimensional (although its units are not stated in the manuscript). This means its value depends on the units of T and c_s . Best to account for this, by perhaps stating that $c_s(z) = k * \text{sqrt}(T)$, where $k=20.05 \text{ ms}^4\text{K}^{1/2}$.**

This has been fixed as suggested by the reviewer.

6. **Line 356: What are the altitude limits on the integrals?**

Limits from 0 to ∞ have been added to the integrals to clarify this is theoretically over the whole depth of the atmosphere - but note the subsequent caveat that in practice we stop at 80 km due to data availability.

This would indeed be fascinating, but sadly the spatiotemporal coverage of the satellite instruments used is not sufficient to observe vertical propagation of the initial large waves directly. This is because the first gravity-wave resolving satellite instrument to pass close enough to observe these waves is ASI-C at around 8:00 UTC, 3.5 hours after the initial eruption (Figure 2). At this time the waves are visible at all heights. This is consistent with a theoretical calculation of the vertical group velocity of waves with the observed properties, which we expect to propagate through the full depth of the atmosphere in less than an hour.

2. The impact of the result on weather and climate forecasting is not well justified. The Tonga eruption is a rare event, and its atmospheric impacts are extreme. It is unclear how the simulation of this extreme event would provide insights into the strengths and deficiencies of models for weather and climate forecasting, which are nearly always used for non-extreme conditions. The authors should clarify the applicability of the knowledge learnt from this extreme event to typical weather and climate forecasting.

We agree with the reviewer comment, and have included our response to this in response to Reviewer

1's second major comment above.

C.2 Minor Comments

1. Lines 46 - 47: The unit of energy is EJ or EJ²?

The "2"s are cross-links to reference number 2, i.e. Pyle et al 2000. This is admittedly a little confusing in the current format of the text! To make this clearer the references have been moved to the text immediately before the bracketed values.

2. Lines 54, "non-acoustic frequencies": what is the range of non-acoustic frequencies? Are the Lamb and gravity waves lower frequency than acoustic waves?

Yes. To make this clearer, "non-acoustic" has been modified to "sub-acoustic".

3. *Caption of Figure 2: “Airglow inser” → “Airglow insert”?*

This was intended to be “inset”, and has been corrected.

4. *Line 78, “shockwave”: where is it and how to determine that it is a shockwave?*

The term ‘shockwave’ was used a little informally here, and should not have been - while other published studies and the media have described the Hunga Tonga Lamb wave as a shockwave, this is not a technically accurate description of the physics involved and the reviewer is correct to flag this up. We have corrected the one instance of this term (line 78) to ‘atmospheric wave’ to avoid any implication of e.g. supersonic physics.

5. *Lines 137 - 138: Extended Data Figure 3 should be Extended Data Figure 6?*

Yes - this has now been corrected.

6. Line 138: where are the “Two low-amplitude wavefronts”? Panels are not labeled in Extended Data Figure 5 (no 5b).

The reference should have been to Extended Data Figure 4b, not 5b. In the correct figure they are indicated by dashed red lines overlaid on the panel.

7. Line 147, “consistent in speed”: please provide explanation. Which two speeds are consistent?

The speed and direction of the wave across the observed portion of New Zealand, if propagated backwards in a Galilean framework, is consistent with an origin at Hunga Tonga in the specified time range, assuming constant speed and direction - i.e. only one speed is involved, not two. To make this clearer, the sentence has been slightly rephrased to “The speed and propagation direction of these waves is consistent with ...”.

8. Line 151, “6.15am”: what does it mean here?

Corrected to 06:15.

9. Lines 221 - 223: any references?

Two references have been added for each comparator.

10. Lines 383-389 and Figure 2 airglow insert: Are the airglow images taken during the night or the day? The text seems to indicate that these are nighttime airglow images, while the figure shows daytime? Can the authors provide the original airglow images?

The airglow observations are nighttime imagery, but with a brighter background than would be typical

due to a full moon on the night of the eruption. The images hence give the impression of daylight at first inspection, but on close examination stars are faintly visible in the far-field. The image shown is the original camera image without any treatment other than the superimposition of dashed lines to highlight the otherwise weakly-visible phase fronts. See also the response to Reviewer 3 on the topic of the airglow images (below), for which we have added some text to the manuscript describing this analysis in greater detail.

11. Extended Data Figure 3: There seems a gap between the TIDs at the west coast of USA and south-eastern USA. Why there is little TID in-between? Does this imply that the TIDs in south-eastern USA are induced by sources other than the eruption?

We have replotted the data in this figure, which resolves this issue. The previous version of the figure considered data separately in the western US and southeastern US and the panel showed a spatial gap between these regions. The new version integrates data across the whole US, and thus the gap is no longer present.

12. Supplementary Figures 1 and 2: it would be better to add timestamps and colorbar.

The figures have been regenerated with colourbars and timestamps.

A Reviewer 2

A.1 Major Comments

1. If possible, I think the authors should take the full advantage of the observational data and investigate the vertical propagation of the waves. This would significantly improve the novelty of the work. The authors have mentioned the vertical wavelength in the manuscript. What about the propagation time? How much time it takes for each type of the waves to reach various observational heights? Does the vertical propagation speed of the wave varies with height? Can all the Lamb and gravity waves reach the ionosphere (possible atmospheric filtering of waves)?

This would indeed be fascinating, but sadly the spatiotemporal coverage of the satellite instruments used is not sufficient to observe vertical propagation of the initial large waves directly. This is because the first gravity-wave resolving satellite instrument to pass close enough to observe these waves is IASI-C at around 8:00 UTC, ~3.5 hours after the initial eruption (Figure 2). At this time the waves are visible at all heights. This is consistent with a theoretical calculation of the vertical group velocity of waves with the observed properties, which we expect to propagate through the full depth of the atmosphere in less than an hour.

2. The impact of the result on weather and climate forecasting is not well justified. The Tonga eruption is a rare event, and its atmospheric impacts are extreme. It is unclear how the simulation of this extreme event would provide insights into the strengths and deficiencies of models for weather and climate forecasting, which are nearly always used for non-extreme conditions. The authors should clarify the applicability of the knowledge learnt from this extreme event to typical weather and climate forecasting.

We agree with the reviewer comment, and have included our response to this in response to Reviewer 1's second major comment above.

A.2 Minor Comments

1. Lines 46 - 47: The unit of energy is EJ or EJ²?

The "2"s are cross-links to reference number 2, i.e. *Pyle et al 2000*. This is admittedly a little confusing in the current format of the text! To make this clearer the references have been moved to the text immediately before the bracketed values.

2. Lines 54, "non-acoustic frequencies": what is the range of non-acoustic frequencies? Are the Lamb and gravity waves lower frequency than acoustic waves?

Yes. To make this clearer, "non-acoustic" has been modified to "sub-acoustic".

3. Caption of Figure 2: "Airglow inser" → "Airglow insert"?

This was intended to be "inset", and has been corrected.

4. Line 78, "shockwave": where is it and how to determine that it is a shockwave?

The term 'shockwave' was used a little informally here, and should not have been - while other published studies and the media have described the Hunga Tonga Lamb wave as a shockwave, this is not a technically accurate description of the physics involved and the reviewer is correct to flag this up. We have corrected the one instance of this term (line 78) to 'atmospheric wave' to avoid any implication of e.g. supersonic physics.

5. Lines 137 - 138: Extended Data Figure 3 should be Extended Data Figure 6?

Yes - this has now been corrected.

6. Line 138: where are the “Two low-amplitude wavefronts”? Panels are not labeled in Extended Data Figure 5 (no 5b).

The reference should have been to Extended Data Figure 4b, not 5b. In the correct figure they are indicated by dashed red lines overlaid on the panel.

7. Line 147, “consistent in speed”: please provide explanation. Which two speeds are consistent?

The speed and direction of the wave across the observed portion of New Zealand, if propagated backwards in a Galilean framework, is consistent with an origin at Hunga Tonga in the specified time range, assuming constant speed and direction - i.e. only one speed is involved, not two., To make this clearer, the sentence has been slightly rephrased to “The speed and propagation direction of these waves is consistent with ...”.

8. Line 151, “6.15am”: what does it mean here?

Corrected to 06:15.

9. Lines 221 - 223: any references?

Two references have been added for each comparator.

10. Lines 383-389 and Figure 2 airglow insert: Are the airglow images taken during the night or the day? The text seems to indicate that these are nighttime airglow images, while the figure shows daytime? Can the authors provide the original airglow images?

The airglow observations are nighttime imagery, but with a brighter background than would be typical due to a full moon on the night of the eruption. The images hence gives the impression of daylight at first inspection, but on close examination stars are faintly visible in the far-field. The image shown is the original camera image without any treatment other than the superimposition of dashed lines to highlight the otherwise weakly-visible phase fronts. See also the response to Reviewer 3 on the topic of the airglow images (below), for which we have added some text to the manuscript describing this analysis in greater detail.

11. Extended Data Figure 3: There seems a gap between the TIDs at the west coast of USA and south-eastern USA. Why there is little TID in-between? Does this imply that the TIDs in south-eastern USA are induced by sources other than the eruption?

We have replotted the data in this figure, which resolves this issue. The previous version of the figure considered data separately in the western US and southeastern US and the panel showed a spatial gap between these regions. The new version integrates data across the whole US, and thus the gap is no longer present.

12. Supplementary Figures 1 and 2: it would be better to add timestamps and colorbar.

The figures have been regenerated with colourbars and timestamps.

D Reviewer 3

L161-181, L202 – although this discussion and explanation can be valid, I would point to the fact that the eruption generated a tsunami. This tsunami could be a source of [GWs] ... and thus 3) various traveling ionospheric disturbances... Although they may be less prominent at far-field in lower atmospheric observations, these waves may potentially be source of detectable fluctuations in near-field, as well as at far-field at ionospheric altitudes. They also demonstrated that refracted/trapped/deflected ocean waves near the shore may drive AGWs/GW for hours.... Signals of 200 m/s looks also comparable with speeds of tsunami propagations (see Figure -2 in the attachments)... To summarize, for the sake of completeness, you may consider adding a tsunami because of undersea eruption and thus AGWs in the atmosphere as sources of detected fluctuations.

The Lamb wave and fast gravity waves associated with the initial eruption had a speed and morphology highly consistent with an atmosphere-only pathway, and travelled substantially faster than the

estimated speed of the tsunami provided by the reviewer. Therefore, we believe it is unlikely that the waves were generated via an oceanic pathway. From the reviewer's comment on the 200 m/s speed of the tsunami wave, we assume that the reviewer agrees with this, but mention it for clarity and completeness.

The slower far-field waves hours after the eruption in many (but not all) cases can be inferred to have speeds of order 200 m/s or lower, and therefore could be triggered by the tsunami. However, we would argue against this on the grounds that the waves we observe are nearly perfectly concentric about Hunga Tonga at all ranges*. An oceanic pathway would be very likely to introduce additional noise to the 'secondary' atmospheric phase fronts produced later - see for example the bottom-topography focusing effects simulated by figure 9 of Inchin et al (2020, doi:10.1029/2020JA028309) and consider also interactions between the Proudman resonance effect and ocean trenches as discussed by Lynett (2022, doi:10.21203/rs.3.rs-1377508/v1) and Tanioka et al (2022, doi:10.21203/rs.3.rs-1320093), which will generate new smaller tsunamis at trench locations. We do not see evidence of these effects in our atmospheric wave measurements. However, we cannot rule this pathway out from our data, and accordingly, we have added a sentence discussing this possibility to the text, and a reference to Inchin et al. We have also added a sentence discussing evidence of meteotsunamis generated in other basins by the atmospheric waves, highlighting the complex ways in which the oceans and atmosphere are coupled by waves.

L32/L65 (and wherever it is mentioned) – The authors may reference, e.g., Liu et al. 1982 who presented results of the detection of ionospheric fluctuation generated by (potentially) Lamb waves after the 1980 Mt. St. Helens eruption and Robert et al (1982), Also, "Ionospheric and atmospheric disturbances around Japan caused by the eruption of Mount Pinatubo on 15 June 1991" by Igarashi et al. 1993.

Thanks! All three references have been added to the manuscript.

For reader, it could be hard to see phases and their continuity to validate reported speed

s

from Extended Data Figure 3. Could some parts of the figure be zoomed in?

We agree that the phase fronts were difficult to resolve in the original version of the figure. This was due to plotting individual measurement points, which had a high level of scatter noise over small distances. To resolve these, we have recomputed the data in one minutes by 5 kilometre bins, which significant clarifies the figure. This binning does also reduce peak magnitudes, and the corresponding values have been changed in the body text to these new magnitudes to avoid confusion for the reader. The raw data used have not changed.

Figure 2 – I understand what is meant by “initial” gravity waves, but I think it is not needed here, as it may potentially be misleading (otherwise introduce what is meant by “initial” and/or subsequent etc.). I would also add panel letters to each observation depicted (i.e., (a) Airglow,

Hawaii, (b) Surface Pressure Stations etc.).

This is a tricky one, as most other words that could be used here could also be potentially misleading (e.g. ‘leading’ could imply some form of guiding, ‘front’ could imply an association with weather, and ‘foremost’ while probably fine for the Lamb wave would change as a function of time for the dispersive gravity waves.). As all three reviewers understood the meaning of ‘initial’ in this context, we have left it unchanged for now, but are very happy to change it if requested.

As suggested, lettering has been added to the panel, and is now used to refer to the individual panels in the text.

How you define the altitude of GNSS observations of 100-250 km? Could the signals be detected and discerned from the noise at such low altitudes as 100 km? Why then the shell height was set at the top boundary of 250 km?

The response of the electron density is a function of the neutral atmospheric perturbation, the geo-

magnetic field angle and the background ionisation and therefore is extended in altitude (see for example simulations by *Bagiya et al* (2019, doi:10.1038/s41598-019-54354-0). Furthermore, *Nicholls et al* (2013, doi:10.1002/2013JA018988) demonstrate from ISR observations examples of the waves extending from 115 to 300 km altitude and this is consistent with previous modelling work (e.g. Hooke papers from the 1960s) that show gravity wave perturbations extending up to the F2-region of the ionosphere.

The GPS signals traverse the entire ionosphere and therefore they will experience the wave in electron density integrated along a line of sight from the satellite to the receiver and observed as fluctuations in the GPS TEC. Only high elevation satellites are used in our analysis to minimise errors in the assumptions necessary for mapping to a specific altitude (i.e. shell height) in the ionosphere. An ionospheric altitude of 250 km is used in our paper, which is commonly selected for TIDs in other works (for example see *Rolland et al*, 2011, doi:10.5047/eps.2011.06.020). To clarify this, we have added *Rolland et al* to the methodological references.

*with the exception of the region immediately west of Hunga Tonga, but this region has significant background GW activity - see ED Figure 9c-e - and hence cannot be used to test this hypothesis

L78 – how to understand, from Figure 1, that visible fluctuations are driven by a shockwave? In the same paragraph, it is mentioned Lamb wave. Not clear what is meant by “mixed packet of waves”. What was mixed? How from Figure 2, Extended Figure 1c,d and Supplementary Figure 1 the reader can find “mixed packet of waves with non-dispersive wavelengths and periods”? What periods and what wavelengths are estimated? What is meant by “large” amplitude? To summarize, it would be great to provide additional quantitative discussion

here and adjust figures accordingly.

The word ‘shockwave’ was poorly chosen and has been replaced in response to a comment from Reviewer 2 - the Lamb wave is the wave in question.

The clause containing the phrase ‘mixed packet’ has been removed from the text as it was unclear. As the amplitude varies substantially by location due to the spreading of the wave, quoting a single value in the text would not be broadly representative, and accordingly we do not do so. The amplitude and temporal period of the Lamb wave at surface level are shown in Figure 1 and ED Figure 1a for the general case and ED Figures 1e and 8 for the specific cases of Broome, Australia and Tonga respectively. The horizontal wavelength is not stated directly, but can be computed from the period and speed of the data as presented. For the stratospheric general case, the amplitude is shown by Figure 2 and ED figure 5b and the horizontal wavelength is shown in ED figure 5c; note that a temporal period cannot be determined from these data as they are pseudo-instantaneous snapshots.

L86 – which panel of Figure 1 demonstrates 4 wavefronts? Can they be marked? I would also suggest adding here panels (a), (b), (c) to direct a reader to exact panels that are mentioned in the text.

The panels have now been lettered, and the relevant panels (m-p) identified in the text. The wavefronts are marked in these panels with black arrows.

Is it possible to provide more insight on the speeds with GOES data? A slowdown of waves over South America is mentioned , but it is not clear if waves were slowed down over South America or earlier, in the Pacific Ocean (not sure I can see it from Extended Data Figure 4,b).

I think time-distance diagram could be useful to support this discussion.

The possible South American slowdown is not easily visible in ED4b as the figure is a radial average and the effect is small and regional. It is also very close to the noise floor of GOES. To better highlight this feature, we have therefore added an additional ED figure (ED 10) of filtered GOES data showing the wave before and after the slowdown. The animated Supplementary Figure 2, where the possible slowdown is seen to take place over the east of South America as a mild bending of the primary phase fronts, also shows this effect as a function of time.

L98 – 308.5 and 319.4 m/s – are these 2 numbers for 2 different locations or 2 different observations?

Different locations, as the text currently says (“a phase speed of between 308.5 and 319.4 ms⁻¹ depending on location”).

L491 – 12300 km
“km” added.

L102 how did you estimate phase speed of 318 m/s for hydroxyl airglow over Hawaii?

There are five visible wavelength cameras at the Gemini observatory in Hawaii, of which the results from one pointing northwards are shown in Figure 2b - results from the others are fully consistent with the chosen image, and accordingly the others are not reproduced both for brevity and clarity. Of the five cameras, one is aimed at a near-vertical angle (with a slight offset determined from study of the star field), and we use this image to identify the arrival time of the first wave packet using the image time stamp - this time is 08:48:53 UTC. At a distance of 4964 km and using an explosion time of 04:28:48 UTC, this gives a phase speed of 318.12 m/s. Further analysis using the other four cameras from the Gemini observatory gives results consistent with this. We use a horizontal view rather than the vertical view as our example in Figure 2b because the vertical image requires enhancement to make the red phase fronts visible, whereas the fully-temporally-consistent horizontal view shows the same signal without the need to modify the image. This information has been added to the methods section.

L103 – what is meant by “uniform” phase fronts? Uniform in what extent? I am not sure that any observations demonstrate vertical structure of fluctuation.

Our observations do provide vertical phase structure over a large chunk of the atmosphere, as described in the text.

To reiterate: for clarity of interpretation, the AIRS, IASI and CrIS data shown in Figure 2 have been shown at a different altitude level for each dataset, but (as described in the caption to Figure 2 and elsewhere) all three instruments each measure the same three stratospheric altitudes (approximately, 25 km, 39 km and 42 km) simultaneously. In each case, these measurements represent an average over a depth approximately 10-12 km centred at the specified height, and due to the strength of the signal and given its brief period it is likely that the phase front is at the same place across the full depth of the measurement. In all three datasets, no apparent phase difference is visible in the Lamb wave at different altitudes.

In addition, the arrival time of the Lamb wave at locations with surface pressure stations is consistent with the arrival time in the stratospheric measurements, as are the airglow images from Hawaii and the lower-troposphere-dominated GOES data. While our measurements do not provide complete coverage over some gaps (approximately 5-15km and 60-80km), based on both the large proportion of the height range covered and the our theoretical understanding of external wave modes, it is very unlikely that phase variations with heights in these parts of the profile are significant, and the vertical phase structure of the wave would have to be very morphologically unusual to only show divergences from uniformity in these altitude ranges.

L107 – Does any of demonstrated instrument or dataset have a sufficient precision to claim that Lamb wave propagated with 319 and 316 m/s in different directions? This is not following from the earlier discussion, where the speeds are suggested with an uncertainty of ± 6 m/s (L81) ± 3 m/s (L85), 5_{\pm} m/s and 4_{\pm} m/s (L98). What are the sampling rates of the data demonstrated?

The reviewer is correct - thank you for spotting this. We have clarified this by changing the text to say that “Our data may show evidence of a slightly different speed for propagation in different directions across the Earth..., but this is within the uncertainty range of our measurements”. For completeness here, we note that strictly speaking the barometer data are not reporting errors but rather genuine variability, and that the uncertainty in the measurements depends on the time sampling of the data (which is variable) and the size of the pressure perturbation (also variable).

L151 – 6.15am

Corrected to 06:15.

L151 (see Figure - 3 in the attachments). Does it mean that TIDs (2) arrived to US later than expected? I am looking at those fluctuations between lines (2) and (3) and distance range

9000-11000 km. If those TIDs arrived later than (2) in New Zealand – are they driven by the same “source” or different? If different, why they are both named as (2)?

Yes, it is correct that the TIDs arrive slightly later than expected theoretically. Given their highly unusual magnitude and the relatively small ‘delay’, we believe this represents delays in propagation over large distances due to e.g. atmospheric refraction rather than a different source mechanism. For this reason, we also assign them the same identifying number.

How TID(4) consistent with eruption? Again, zooming of regions of discussion could be helpful. Just an idea for the figure - see Figure - 4 in the attachments (ignore if you don’t like it). Meaning to put a pressure plot to the bottom of Extended Figure 3. From first try, the TIDs 4 is kind of consistent with 4th explosion, but not fully. By saying “TIDs 4 are consistent with the wave activity generated over Hunga Tonga in the hours after the primary eruption” – do you mean that TIDs 4 is from Lamb waves generated during explosion 4? I think some clarifications can be provided here.

This is a very useful suggestion that has significantly enhanced our analysis shown in ED 3. We have now added the Tonga pressure signal to the figure as an additional panel, aligned to correspond temporally. This addition shows clearly that TID 4 is temporarily highly consistent with a pressure drop seen at Tonga possibly corresponding to a later smaller eruption. We have adjusted the text to mention this.

E Other Changes

References 8 and 17 (in the original numbering) were preprints that have now been accepted and published. We have updated these to their as-published designations. In addition, we have fixed a small number of typographical errors from the original beyond those identified by the Reviewers, and added a funder acknowledgement which was accidentally omitted from the original submission.

Reviewer Reports on the First Revision:

Referees' comments:

Referee #1 (Remarks to the Author):

The authors have made appropriate revisions to respond to my previous comments. The use of statistics is appropriate. I have no further comments and am happy to recommend the manuscript for publication.

Referee #2 (Remarks to the Author):

The authors have satisfactorily addressed the comments made by the reviewers and presented a significantly improved manuscript. I only have two minor comments on "Weather and Climate Forecasting Implications" section:

1. I wonder if the authors could clarify the altitude coverage of "atmospheric models"? I assume these were whole atmospheric models like WACCM that covers all the way up to the mesosphere and above? The authors mention "weather models" in the second to the last paragraph. Would the "weather" models refer to those resolving the troposphere only?

2. For the last paragraph, if I understand it correctly, comparing the modeled and observed propagation speeds would help evaluate how well models represent the background atmosphere (including winds, temperatures, density structures), through which the waves propagate. Is there any direct observations of the background atmospheric conditions so the modeled background atmospheric conditions could be compared with? If not, then it makes sense to evaluate the model performance via comparing modeled and observed wave propagation speeds.

Referee #3 (Remarks to the Author):

The manuscript was substantially improved, now providing representative figures, references to them and necessary discussion. I believe it can be considered for the publication at this stage. Below I provide a couple of minor comments to the comments.

1. Although I think that the discussion of tsunami-related AGWs is sufficient now, I would like to note that tsunamis drive packet of AGWs of a broad range of periods and spatial scales, with faster propagating phases at the head of the packet (longer periods/horizontal wavelengths) to slower propagating short period AGWs at the tail of the packet, thus not necessarily strictly freezing around a dominant tsunami period (same Inchin et al. 2021; Vadas et al. 2015 etc.). Mentioned in the revision Bagiya et al. demonstrated observations and modeling of AGWs from tsunami that propagate with much faster horizontal velocities than tsunami (<https://doi.org/10.1002/2017JA023971>, and references therein). However, I don't think that it is

possible to clarify this in the framework of the current study that targets observational part of the problem.

2. I found interesting discussion in the revision on modeling capabilities. I'd like to add that although modern NWP may be hard to use in simulating such dynamics (very close though), wave propagation models, such as used in papers by Fritts, Dong, Inchin, Heale, etc. allow simulating AGW propagation over large areas from ground to the ionosphere with resolutions <km (yet expensive!).

3. Thanks for the clarification with delays over CONUS, very nice animations. For authors' consideration, maybe 1-2 sentences on reasons for delays could be useful. Recently, <https://doi.org/10.1029/2022GL098324> also discussed this based on Himawari-8 satellite observations. In the Discussion section, I found several useful references related to the investigation of the Lamb wave propagation, especially in non-isothermal atmosphere.

Response to Reviewers

A Reviewer Comments

As the Reviewer Comments in this revision cycle are very brief, we have combined them into a single-section response. Note that Reviewer 1 had no additional comments on our manuscript beyond recommending it for publication.

Reviewer 2 - Comment 1: *I wonder if the authors could clarify the altitude coverage of "atmospheric models"? I assume these were whole atmospheric models like WACCM that covers all the way up to the mesosphere and above? The authors mention "weather models" in the second to the last paragraph. Would the "weather" models refer to those resolving the troposphere only?*

To clarify that this point is intended generally, we have added the text "operating across all levels of the atmospheric system" to the end of this sentence.

Reviewer 2 - Comment 2. *For the last paragraph, if I understand it correctly, comparing the modeled and observed propagation speeds would help evaluate how well models represent the background atmosphere (including winds, temperatures, density structures), through which the waves propagate. Is there any direct observations of the background atmospheric conditions so the modeled background atmospheric conditions could be compared with? If not, then it makes sense to evaluate the model performance via comparing modeled and observed wave propagation speeds.*

The paragraph is intended to highlight the latter possibility, i.e. that model performance can be evaluated by comparing modelled and observed wave propagation speeds. We have modified the text slightly to make this clearer.

Reviewer 3 - Comment 1: *Although I think that the discussion of tsunami-related AGWs is sufficient now, I would like to note that tsunamis drive packet of AGWs of a broad range of periods and spatial scales, with faster propagating phases at the head of the packet (longer periods/horizontal wavelengths) to slower propagating short period AGWs at the tail of the packet... However, I don't think that it is possible to clarify this in the framework of the current study that targets observational part of the problem.*

We agree with the reviewer that, while this could be an interesting avenue of research, it is significantly beyond the framework of the current study.

Reviewer 3 - Comment 2: *I found interesting discussion in the revision on modeling capabilities. I'd like to add that although modern NWP may be hard to use in simulating such dynamics (very close though), wave propagation models, such as used in papers by Fritts, Dong, Inchin, Heale, etc. allow simulating AGW propagation over large areas from ground to the ionosphere with resolutions <km (yet expensive!).*

We agree with the reviewer, and reference 42 (Fritts et al, JGR 2022) describes such a model.

Reviewer 3 - Comment 3: *Thanks for the clarification with delays over CONUS, very nice animations. For authors' consideration, maybe 1-2 sentences on reasons for delays could be useful. Recently, <https://doi.org/10.1029/2022GL098324> also discussed this based on Himawari-8 satellite observations. In the Discussion section, I found several useful references related to the investigation of the Lamb wave propagation, especially in non-isothermal atmosphere.*

We are slightly confused by this comment, as the paper does not refer anywhere to delays over CONUS (which we assume expands out to 'Continental United States', as this is the top result for an acronym search for this term). From broader context, we assume that the Reviewer is referring instead to the delays seen over northern South America, which we do discuss and show an animation of, and which Reviewer 3 had comments on in the first cycle of revisions.

While we again agree that this would be interesting, it is again slightly beyond scope as the problem is quite complex - for example, at time of writing, personal communications by the lead author with

other research groups suggest that modellers are still having trouble reproducing this effect in dedicated simulations of the Hunga Tonga Lamb wave. The references in the Discussion of the paper suggested by the Reviewer (Otsuka, GRL 2022) do provide some hypotheses, but simply reproducing this information in a very compressed sentence or two here would not add significantly to the work we report, particularly since given the large size of our figures we estimate that we are currently very close to the page limit for this manuscript and would need to add a large number of references hereto do this topic justice.